# Spectral Domain Neural Reconstruction for Passband FMCW Radars

## Abstract

We present SpINR, a novel neural framework for high-fidelity volumetric reconstruction of small, near-field tabletop objects using Frequency-Modulated Continuous-Wave (FMCW) radar. Traditional radar imaging techniques often apply FFT as a black-box post-processing step, discard phase information, and require dense aperture sampling, leading to limitations in sub-centimeter reconstruction and generalization. Our core contribution is a fully differentiable frequency-domain forward model that captures the complex radar response using closed-form synthesis, paired with an implicit neural representation (INR) for continuous volumetric scene modeling. Unlike time-domain baselines and magnitude-only spectral pipelines, SpINR directly supervises the complex frequency spectrum from raw 1D chirps acquired along an arbitrary cylindrical aperture, preserving phase-sensitive spectral spectral fidelity while drastically reducing computational overhead. Additionally, we introduce sparsity and smoothness regularization to disambiguate sub-bin ambiguities that arise at high carrier frequencies and fine range resolutions. Experimental results show that SpINR significantly outperforms both classical and learning-based baselines, with a 52.6% improvement in reconstruction quality and 32% improvement in latency.

## 1 Introduction

Volumetric scene perception has become increasingly vital across a range of modern applications—from autonomous navigation and augmented reality to robotics, healthcare imaging, and activity monitoring. These applications increasingly rely on accurate reconstruction of 3D geometry from sparse, distributed measurements, prompting the development of sensing modalities and algorithmic frameworks. While much of the recent literature emphasizes large-scale indoor or urban environments, many emerging applications (e.g., tabletop manipulation, fine-grained inspection, and small-object tracking) require *high-fidelity reconstruction of small, near-field volumes,* where wavelength and geometric detail are of comparable scale.

In recent years, Frequency-Modulated Continuous-Wave (FMCW) radar has gained popularity for volumetric perception in new applications. Its hardware-efficient architecture enables fine-grained range estimation, while its frequency-domain formulation allows compact signal processing pipelines amenable to embedded deployment. Naturally, FMCW radar can benefit from emerging neural volumetric imaging methods for high-resolution scene reconstruction. However, existing neural methods are typically built around 2D range–azimuth (or range–Doppler) tensors from planar MIMO arrays under far-field assumptions, and often struggle to model the unique spectral properties of FMCW signals in the *small, near-field tabletop regime.* In particular, they rarely account for the complex passband phase and Dirichlet-kernel spectral leakage that arise when reconstructing fine detail from raw 1D chirps collected along non-planar, sparse synthetic apertures, motivating the need for new approaches that align with the physics of beat-frequency formation while remaining computationally tractable.

To this end, we present **SpINR** – an approach that combines *physically-grounded signal modeling* with the expressive power of *implicit neural representations (INRs)*. The key insight is to operate *directly in the frequency domain* for FMCW radars, where range is linearly encoded in beat frequency, enabling a *closed-form differentiable forward model*. This allows us to synthesize only the frequency components relevant to the scene's spatial extent—unlike time-domain models that simulate redundant data or 2D range–azimuth pipelines that impose mismatched far-field assumptions.

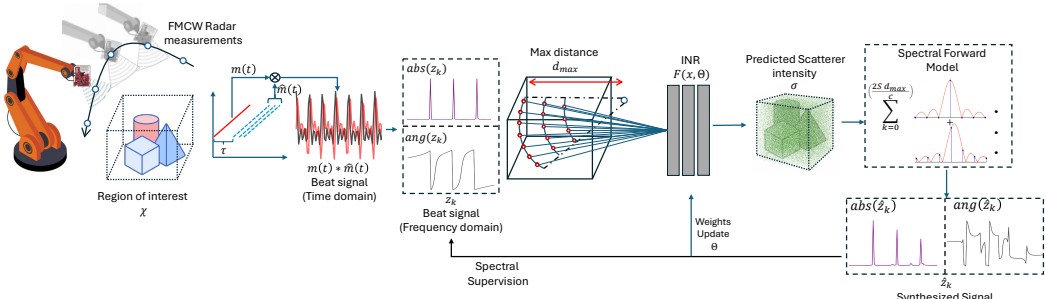

Figure 1: Overview of SpINR. SpINR proposes a spectral forward model and spectral supervision for FMCW signal synthesis.

SpINR's contributions can be summarized as follows.

- **Spectral supervision with differentiable forward model** – we derive a fully differentiable frequency-domain forward operator for FMCW radar that analytically models spectral leakage and passband phase, and we supervise the implicit neural representation directly in the complex spectrum, avoiding time-domain instability and magnitude-only approximations.

- **Scalability and efficiency** – the proposed forward model and sampling strategy enables selective frequency bins synthesis relevant to the bounded tabletop volume, and are designed to operate on raw 1D chirps from sparse cylindrical apertures, without requiring dense planar MIMO arrays or 2D range–azimuth tensors.

- **Frequency-selective spectral synthesis** – by working natively in the frequency domain, SpINR only synthesizes those DFT bins whose ranges overlap the bounded scene volume, reducing computation and improving latency by 32%.

- **Novel regularizations and sampling strategies** – to extend the modeling to higher frequencies, where shorter wavelengths introduce sub-bin ambiguity, we introduce novel *regularizations* and *sampling strategies*.

We validate SpINR through extensive experiments and ablations on synthetic but geometrically complex tabletop scenes (e.g., standard benchmark objects with fine, non-convex structure), showing that it outperforms classical backprojection, range-quantization models, time-domain baselines, and prior learning-based models across geometric and perceptual metrics. Our work is complementary to recent neural radar methods on large outdoor/indoor environments, and sets a new precedent for neural radar reconstruction in small, near-field, high-frequency regimes where prior methods degrade significantly.

## 2   RELATED WORK

**FMCW Radar Imaging and Signal Modeling.**   FMCW radar systems offer a lightweight, energy-efficient modality for range estimation, particularly well-suited for embedded applications in robotics and autonomous driving. The radar's *beat frequency encodes distance*, making it amenable to spectral methods like FFT for range estimation. Classical approaches for volumetric reconstruction such as backprojection Duersch (2013), range-Doppler processing Wagner et al. (2013), and SAR Moreira et al. (2013) assume dense aperture sampling, ideal propagation, and often planar array or far-field conditions, making them less robust to real-world imperfections and ill-suited for small, near-field tabletop volumes. Recent learning-based efforts in radar imaging have largely focused on *time-domain models* and coarse voxelized reconstructions Xu et al. (2022); Sonny et al. (2024). However, these neglect key spectral properties like *bin mismatch*, *phase spillage*, and *DFT leakage* Lyon (2009), which are essential for high-precision reconstruction. SpINR differs fundamentally by building a *native frequency-domain model*, capturing radar physics in a closed-form formulation and directly optimizing over spectral observations.

**Learning-Based Radar Reconstruction.**   Deep learning has recently been applied to radar-based depth estimation Lo & Vandewalle (2021), occupancy prediction Ronecker et al. (2024); Borts et al. (2024a), and neural beamforming Al Kassir et al. (2022). A line of work on 2D radar images (e.g., RadarFields and DART ?Huang et al. (2024)) learns neural fields over 2D range–azimuth (or range–Doppler) tensors produced by planar MIMO arrays, demonstrating impressive results on large-scale

indoor and urban scenes under far-field assumptions. These approaches are complementary to ours: they operate on post-processed 2D radar images and coarse spatial resolution, whereas SpINR targets *high-fidelity reconstruction of small, near-field tabletop volumes* directly from raw 1D chirps acquired along a cylindrical synthetic aperture, where fine geometric detail and phase-sensitive spectral effects dominate. More generally, many existing learning-based approaches still operate in the *time domain* or on magnitude-only spectral maps and ignore the full complex spectral structure, limiting their ability to scale to high-resolution volumetric reconstruction. Our approach goes further by combining *complex spectral supervision* with *INRs* and an analytic FMCW forward model, resulting in higher geometric and perceptual fidelity in this small-volume, high-frequency regime.

**Implicit Neural Representations.** INRs like NeRF Mildenhall et al. (2021) represent 3D scenes as continuous functions, mapping coordinates to intensity or radiance. While these models have revolutionized image-based view synthesis, recent extensions have explored their use in medical imaging Chu et al. (2024), audio Wysocki et al. (2024), and transient signals Malik et al. (2023) as input to these models for volumetric reconstruction. Our work brings this paradigm into the *FMCW radar domain*, modeling a continuous radar reflectivity field in the frequency domain using a differentiable forward model. To our knowledge, SpINR is the first to model complex small-scale neural volumes using magnitude and phase of high frequency mmWave FMCW radars. We achieve this by integrating a *spectral-aware, closed-form FMCW forward operator* with an INR for radar signal reconstruction, supervising the network directly on complex spectra rather than on time-domain signals or magnitude-only range images.

**MIMO Radar and Synthetic Aperture Systems.** Multi-Input Multi-Output (MIMO) radar Li & Stoica (2008) and synthetic aperture setups improve angular and spatial resolution by synthesizing dense virtual arrays. Prior work Yanik et al. (2020) processes MIMO data using matched filtering or voxelized accumulation, typically assuming uniform planar sampling and far-field or paraxial propagation. In contrast, SpINR supports *non-uniform cylindrical sampling* and works directly with monostatic measurements at arbitrary poses—leveraging both diversity and sparsity in a differentiable pipeline tailored to small, near-field volumes, without requiring the construction of intermediate 2D range–azimuth tensors.

## 3 PRIMERS AND CHALLENGES

■ **FMCW over Pulse-based Ranging:** Active volumetric imaging techniques fundamentally rely on ranging, i.e., estimating distance from propagated signals. Modulated waveforms emitted by a coherent source reflect off points in the scene and are received as superpositions of time-delayed copies at the receiver. Consequently, the choice of ranging modulation and its corresponding forward model plays a central role in volumetric reconstruction. Early neural volumetric reconstruction methods Reed et al. (2023) primarily considered pulse–echo time-of-flight systems, where transducers are assumed to emit short pulse-shaped signals for imaging. While pulse-based forward models are mathematically simple to formulate, they are not effective for a large and practically important family of imaging radars that employ FMCW signals.

Pulse-based ranging Sarbolandi et al. (2018) offers clear advantages for long-range applications with ample infrastructure and power, such as weather forecasting and air-traffic monitoring. In contrast, many ubiquitous perception applications demand short-range, high-resolution, and energy-efficient imaging radar systems in a compact form factor. This requirement has driven the recent adoption of FMCW imaging radars, which provide low peak power and high SNR at short range. Several off-the-shelf devices, such as TI mmWave radars, integrate high-bandwidth FMCW transceivers on a single chip. As a result, approximately 38% of modern radar devices employ FMCW signal modulation. It is therefore natural to seek ways for FMCW radar to benefit from emerging neural volumetric imaging techniques for high-resolution scene reconstruction. However, directly applying these methods to FMCW radars is non-trivial due to several unique challenges. We elaborate on two major difficulties in the remainder of this section, following a brief introduction to the essentials of FMCW formulation.

■ **FMCW Chirp and Beat Signal:** FMCW uses frequency modulated signal, called chirp, containing a tone with periodically increasing and decreasing frequency. SpINR is developed based on the 'sawtooth chirp', commonly used in imaging and automotive radars where the frequency sweeps up from starting frequency $f0$ with slope $S$ within a time period of $T_c$. The FMCW

chirp, $m(t) = e^{(j2\pi(f_0t+0.5St^2))}$, reflects off a point scatterer in the scene. The reflected signal is scaled by the path loss factor of the signal intensity $\sigma$ and delayed by the round trip time of the signal $\tau$. The FMCW radar sensor 'dechirps' the received signal, $r(t) = \sigma m(t - \tau)$, by mixing it with the transmitted version to obtain a measurement signal called the *beat signal*, $b(t) = r(t)m^*(t) = e^{(j2\pi f_0\tau+St\tau-0.5St^2)}$. The $-0.5St^2$ term in the above equation introduces a negligible transient phase Wang et al. (2014) in the beat signal and often ignored to simplify the equation to $b(t) = e^{j2\pi f_0\tau} \cdot e^{j2\pi(S\tau)t}$. The second exponential in this equation indicates that $b(t)$ is a periodic signal with frequency $(S\tau)$.

**Range from beat signal.** The beat signal essentially transforms the time delay, $\tau$, in the reflected FMCW chirp into a unique beat frequency, $f_b = S\tau$. The beat frequency corresponds to the distance of the scatterer, or rather the total distance the signal travels from the transmitter to the reflecting point and back to the receiver. Transforming the beat signal, $b(t)$ to spectral domain using Fast Fourier Transform (FFT) reveals the beat frequency, $f_b$, and it is mapped to a distance. The range resolution or the radar's ability to distinguish between two nearby scatterers in depth is, therefore, determined by the frequency resolution of the FFT $\Delta f = \frac{1}{T_c}$, considering FFT time window matches the time duration of the chirp. Using the beat frequency relationship, we can represent this frequency resolution to distance resolution as $\Delta d = \frac{c}{2B}$, where $c$ is the speed of the wave in the given medium. As the distance resolution indicates, a wider bandwidth yields finer depth resolution.

■ **Intuitions and Challenges with Beat Signal:** The formulation above considered a single scatterer for simplicity. In practice, a scene contains many scatterers whose time-delayed returns superimpose in the measured signal according to the forward model. The goal of volumetric imaging is to solve the corresponding inverse problem and recover the underlying 3D structure.

In pulse-based ranging, the echo signal is a train of delayed impulses that directly encodes scatterer locations in time. By contrast, the FMCW beat signal in the time domain is a noisy-looking mixture of oscillatory components with no obvious correspondence to spatial structure. Scatterer locations instead appear as distinct beat frequencies in the spectrum, although the mapping from distance to a single DFT bin is only approximate due to spectral leakage into neighboring bins. Thus, learning-based FMCW imaging naturally calls for frequency-domain modeling, but this introduces specific challenges for neural reconstruction.

**Challenge-A: Time-domain Beat Signal is Ill-conditioned.** At the core, neural radar reconstruction methods follow the standard analysis-by-synthesis approach that uses a forward model to simulate the time-domain beat signal for considered scatterers in each iteration and trains a network with supervision using an MSE loss. Contemporary neural techniques primarily retains traditional backprojection's Duersch (2013) approach to use Time-domain Forward model (TF) and Temporal Supervision (TS) – we refer to as TF-TS methods. Temporal supervision on FMCW beat signal, however, does not maintain a simple relationship with the scatterer location.

As explained above, the beat signal in the time domain is a collection of rapidly oscillating sinusoids and any small shifts in the scatterer distance translate to phase shifts in the waveform—causing large changes in MSE, instead of a smooth and convex function to follow toward an optimal solution. This instability makes optimization difficult and prevents the model from learning a meaningful reconstruction and fast convergence. To circumvent this challenge, an intuitive update can be applying supervision in the frequency domain, since the beat signal maps the scatterer distance linearly to beat frequencies. This leads to the family of possible neural reconstruction approach, we call 'Time-domain Forward model with Spectral Supervision' or TF-SS where we first synthesize the full signal in time-domain, then apply an FFT on it for spectral domain transformation, and finally compute loss on the spectral bins.

**Challenge-B: Time-domain Forward modeling is Inefficient.** TF-SS provides spectral supervision, which is certainly stable for FMCW beat signals. However, it still needs to simulate all time domain samples for the entire time period of the FMCW chirp $T_c$ per scatter and perform a full FFT per iteration – while only a small number of corresponding FFT coefficients or 'frequency bins' are useful in the scene analysis. For instance, in a typical mmWave radar system, $T_c$ can lead to 256 samples for an acceptable FFT resolution, however considering a maximum range of the scene of a few meters only the first 16 frequency coefficients covers for maximum possible beat frequency corresponding to the maximum round trip delay. Naive spectral modeling leads to prohibitive computational overheads that grows with the number of scatterers considered for high-resolution scene

reconstruction. A simpler but flawed alternative is to directly assign a scatterer's contribution to a frequency bin based on its range—what we term range-quantized supervision. However, it leads to poor reconstruction as the approximation neglects two critical signal properties: (i) the complex-valued nature of FMCW signals, where both phase and amplitude must be modeled, and (ii) spectral leakage, where energy smears into neighboring bins unless the delay perfectly aligns with a bin center.

SpINR addresses these issues by a novel approach of formulating the entire analysis-by-synthesis pipeline in frequency domain, including a detailed closed-form forward model. Our approach enables precise and efficient supervision on only the relevant spectral components, balancing physical accuracy with computational efficiency.

## 4    FORMULATION OF SpINR

■ **Spectral Transformation:** Spectral analysis of sampled time-series data is performed via a domain transformation of the signal using the Discrete Fourier Transform (DFT). The process presents the frequency spectrum discretized into uniformly spaced frequency bins with representative angular frequencies $\beta_k = \frac{2\pi k}{N}$, where $k$ is the bin index. The Fourier coefficient for each bin represents the phase and amplitude information for the corresponding frequency.

To understand how the DFT distributes energy across bins, consider a complex exponential signal $S_t = Me^{i(\alpha t + \phi)}$ with angular frequency $\alpha$ in radians per sample and sampling step $\Delta t = T_c/N$. The DFT coefficient $Z_k$ evaluates to:

$$Z_k = \frac{M}{N} e^{i\phi} \sum_{t=0}^{N-1} e^{i(\alpha - \beta_k)t} = \frac{M}{N} e^{i\phi} \cdot \frac{1 - e^{i(\alpha - \beta_k)N}}{1 - e^{i(\alpha - \beta_k)}}$$

Considering $\beta_k N = 2\pi k$, the above expression can be rewritten in closed form as:

$$\boxed{Z_k = \frac{M}{N} e^{i\phi} \cdot \frac{1 - e^{i\alpha N}}{1 - e^{i(\alpha - \beta_k)}}} \tag{1}$$

Given that the discrete bin frequencies $\beta_k$ are orthogonal to each other, when $\alpha$ matches a bin frequency, the DFT produces a non-zero coefficient for bin $k$ and (ideally) zero for other bins. However, as Eq. equation 1 shows, when $\alpha$ does not exactly equal any bin, the closest bin in frequency attains the largest coefficient, but all other bins also receive non-zero energy. This phenomenon is known as spectral leakage and is often not modeled explicitly. The closed-form, per-bin expression in Eq. equation 1 is the building block for SpINR's *frequency-selective spectral synthesis*: it allows us to analytically synthesize only the small subset of DFT bins whose range support overlaps the bounded tabletop volume, rather than the full spectrum. The extent of this leakage in neural volumetric reconstruction is explored next.

**Spectral Leakage.** The frequencies in the beat signal represents the distance of the scatters in the scene. If scatterers lies at distances that their corresponding beat frequencies are aligned perfectly to the the bins' center frequencies, all signal energy is concentrated in discrete bins. It results in a clean and unambiguous spectral response. Certainly it is a hypothetical scenario. In practice, distances of a scattering point is arbitrary and therefore the resulting beat frequency is often off the bin center. In such cases, the signal's energy does not remain localized but spreads across multiple neighboring bins and this spectral leakage can be modeled by the Dirichlet kernel: $|Z_k| \propto \left| \frac{\sin\left(\frac{N}{2}(\alpha - \beta_k)\right)}{\sin\left(\frac{1}{2}(\alpha - \beta_k)\right)} \right|$. This results in a sinc-like envelope centered at the true frequency, with energy distributed into adjacent bins. This introduces both amplitude and phase spillage across the spectrum. Importantly, this effect must be explicitly modeled in the forward process. Simplified approaches, such as range quantization, ignore this leakage and instead assign all energy to the nearest bin. Such approximations fail to capture the true complex domain spectral behavior of the signal and degrade reconstruction accuracy. This spectral leakage is a direct consequence of using finite-length DFTs and becomes especially critical in radar signal processing, where accurate frequency estimation directly impacts range resolution and reconstruction fidelity.

■ **Forward Model with Spectral Synthesis:**

In this section, we define the frequency-domain forward model that serves as the basis for our implicit neural representation framework for volumetric reconstruction. Instead of modeling the time-domain signal and applying a DFT afterward, we directly model the complex frequency-domain response at each DFT bin, leveraging the closed-form DFT derivation. This formulation is fully differentiable and thus amenable to gradient-based optimization used in neural rendering pipelines. Moreover, it encapsulates physical wave propagation, including transmission and backscattering, within the radar's operating bandwidth.

Let $\mathbf{x} \in \mathbb{R}^3$ denote a 3D point in the scene, and let $\sigma(\mathbf{x}) \in \mathbb{R}$ represent the reflectivity or scattering strength at that point. Define: $\mathcal{X} \subset \mathbb{R}^3$: the bounded spatial domain of interest, $\mathbf{o}_T, \mathbf{o}_R \in \mathbb{R}^3$: positions of the transmitter and receiver, $R_T(\mathbf{x}) = \|\mathbf{o}_T - \mathbf{x}\|$: propagation distance from Tx to $\mathbf{x}$, $R_R(\mathbf{x}) = \|\mathbf{o}_R - \mathbf{x}\|$: propagation distance from $\mathbf{x}$ to Rx, $\tau(\mathbf{x}) = (R_T(\mathbf{x}) + R_R(\mathbf{x}))/c$: total round-trip delay.

Then, the complex response at DFT bin index $k$ is:

$$Z_k = \int_{\mathcal{X}} \frac{\sigma(\mathbf{x})}{R_T(\mathbf{x}) R_R(\mathbf{x})} \cdot e^{i\, 2\pi f_0 \tau(\mathbf{x})} \cdot \frac{1 - e^{i(\alpha(\mathbf{x}) - \beta_k)N}}{N\left[1 - e^{i(\alpha(\mathbf{x}) - \beta_k)}\right]}\, d\mathbf{x}$$

where $\alpha(\mathbf{x}) = 2\pi S\, \tau(\mathbf{x})\, \Delta t$ is the beat angular frequency *per sample*, $\beta_k = \frac{2\pi k}{N}$ is the angular frequency of the $k$-th DFT bin, $e^{i\, 2\pi f_0 \tau(\mathbf{x})}$ is the *passband* carrier-phase term, $S$ is the chirp slope ($S = B/T_c$), $N$ is the number of time-domain samples (or DFT points). Each beat-signal DFT bin $Z_k$ is thus expressed as an integral over the scene with a complex-valued kernel that encodes the FMCW modulation, the finite-time observation window (via the Dirichlet kernel governing off-bin spectral leakage), and the passband phase induced by the round-trip delay and chirp slope. Crucially, this closed-form spectral operator takes the frequency index $k$ as an explicit parameter, so that we can evaluate it only on the subset of DFT bins whose ranges intersect the bounded tabletop volume, avoiding explicit time-domain synthesis and full FFT evaluation. This forward model represents the coherent summation of returns from all scene points, modulated by the reflectivity $\sigma(\mathbf{x})$, and the frequency response derived from the DFT of a complex exponential tone with a delay $\tau(\mathbf{x})$. Given ground truth measurements $\tilde{Z}_k$, the loss is defined as:

$$\mathcal{L} = \underbrace{\sum_k \left\| |Z_k| - \left| \tilde{Z}_k \right| \right\|_2^2}_{\mathcal{L}_{\text{mag}}} + \lambda_{phase} \underbrace{\sum_k \left( \left\| \Re(Z_k) - \Re(\tilde{Z}_k) \right\|_2^2 + \left\| \Im(Z_k) - \Im(\tilde{Z}_k) \right\|_2^2 \right)}_{\mathcal{L}_{\text{phase}}} \quad (2)$$

where $\lambda_{phase} = 0.5$ controls the relative importance of magnitude vs. complex component supervision.

This formulation can be extended to incorporate additional wave propagation phenomena, such as transmission attenuation models (e.g., based on material absorption), scattering probability as a function of incident angle or local geometry (e.g., Lambertian or specular models), and multipath interference. These extensions can be embedded as additional multiplicative factors within the integrand. Equivalently, the reflectivity field can be generalized from a purely positional term $\omega(\mathbf{x})$ to a direction-dependent field $\omega(\mathbf{x}, \mathbf{u})$, where $\mathbf{u}$ encodes the local incidence/view direction and can include standard radar directivity patterns. In this work, we instantiate an *isotropic* reflectivity $\omega(\mathbf{x})$ that depends only on position, which is sufficient for our small, near-field tabletop regime; incorporating explicit direction-dependent scattering and multipath is orthogonal to our contribution.

■ **Regularizations:** Although our frequency-domain forward model captures the spectral scene response analytically, it suffers from sub-bin ambiguities—particularly for smaller wavelength $\lambda$ when $\lambda/4 < \frac{c}{2B}$. In this regime, different spatial configurations can produce identical phase responses within a single bin, making optimization unstable. Additionally, higher start frequencies introduce global phase offsets that further exacerbate this ambiguity. To address these issues, we introduce:

**(1) Smoothness Regularization.** We encourage local spatial continuity in the predicted scattering field $\sigma(\mathbf{x})$ by penalizing its variation under small perturbations: $\mathcal{L}_{\text{smooth}} = \mathbb{E}_{\delta\mathbf{x}} \|\sigma(\mathbf{x}) - \sigma(\mathbf{x} + \delta\mathbf{x})\|_1$, where $\delta\mathbf{x} \sim \mathcal{U}(-\epsilon, \epsilon)^3$ models small spatial offsets.

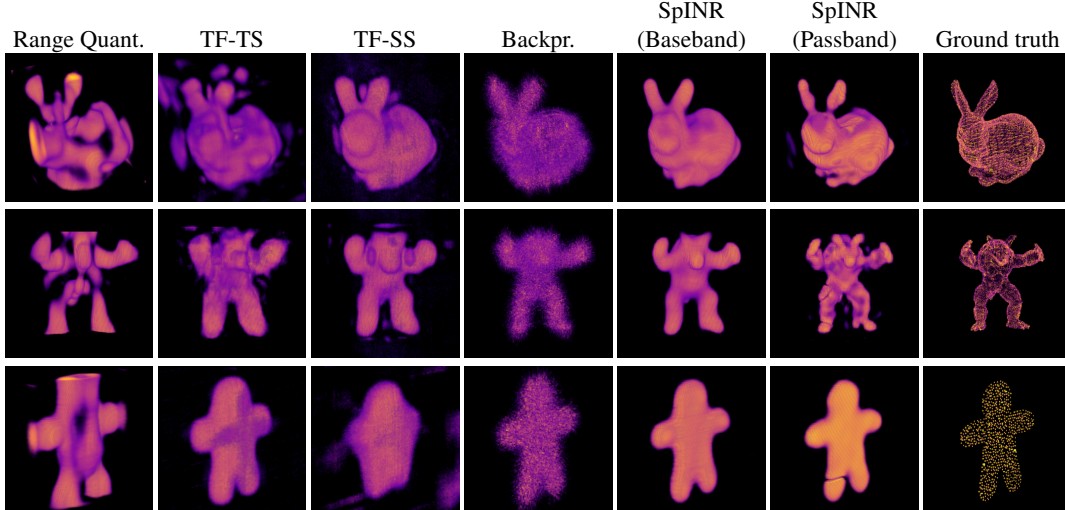

Figure 2: Comparison of volumetric reconstructions for RQ, TF-TS, TF-SS, Backprojection , and SpINR.

**(2) Sparsity Regularization.** Since most of the scanned volume is empty, we enforce sparsity in the scattering field by minimizing its $\ell_1$-norm: $\mathcal{L}_{\text{sparsity}} = \mathbb{E}_{\mathbf{x}} \, |\sigma(\mathbf{x})|$ . This term biases the model toward compact reconstructions and suppresses ghost scatterers. Our final loss becomes: $\mathcal{L}_{\text{total}} = \mathcal{L}_{\text{spectral}} + \beta \mathcal{L}_{\text{smooth}} + \gamma \mathcal{L}_{\text{sparsity}}$, where $\beta$ and $\gamma$ are scalar hyperparameters.

■ **Scene Sampling:** Training the neural reconstruction network requires synthesis of the signal at a specific sensor location for a set of sample scatterer locations representative of the the scene. These scatterer locations are randomly sampled rather than using uniform grid-based approaches common in classical signal processing methods. Random sampling is essential for INRs to enable the network to learn generalizable continuous representations across the entire scene avoiding biases for grid-aligned features and aliasing artifacts Najaf & Ongie (2024). Existing works simulate random ray directions within the beamwidth of the sensor and then sampling points on the rays at uniform depths with respect to the source. We consider cylindrical aperture for sensor locations – a common approach for volumetric scene analysis. In this case, we observe that uniform depth-based sampling leads to systematic undersampling of the scene center and superfluous sample density as we move away from the central axis. To maintain uniform point density, we introduce cubic sampling. We draw depths $r$ with a probability density that compensates for the quadratically growing cross-sectional area of the cone. A uniform random variable $u \sim \mathcal{U}(0,1)$ is mapped to the desired radial distance via $r(u) = \left( u\left(r_{\text{far}}^3 - r_{\text{near}}^3\right) + r_{\text{near}}^3 \right)^{1/3}$. Additionally, for cases where scene dimensions are known beforehand, samples can also be drawn randomly within scene boundaries to minimize the number of sampling points. We evaluate these three sampling methods in our evaluation.

## 5 EVALUATION

■ **Experimental Setup:** We evaluate SpINR using simulated FMCW radar data over a cylindrical synthetic aperture. The setup mimics a practical configuration where the object rests on a turntable and the radar sensor is mounted on a vertical actuator. Their combined motion results in a cylindrical inverse synthetic aperture. Our radar simulation follows the commercial TI AWR1843BOOST MIMO configuration, with a 3.585 GHz bandwidth, $70.295 \times 10^{12}$ Hz/s chirp slope, and a sampling rate of 5 MHz. We define two variants of SpINR - SpINR (Passband), where $f_0 = 5 GHz$, and SpINR (Baseband), where $f_0 = 0$. In all experiments we use $N = 256$ time samples per chirp and a 256-point DFT (full-chirp rectangular window), matching the AWR1843BOOST-like frontend. The scene spans a 0.36 m cube, with the radar positioned 0.23 m from the scene centre. This sampling strategy closely mirrors real-world volumetric setups such as AirSAS Cowen et al. (2021), and provides a feasible path toward hardware realization compared to more idealized setups like spherical apertures. We train our models on one RTXA5000 GPU and use 50% randomly selected

Table 1: Reconstruction metrics across all scenes and Chamfer distance (cm) for individual scene.

| Method | Metrics (Across Scenes) | | | | | Chamfer distance across Scenes | | | | | | |
|---|---|---|---|---|---|---|---|---|---|---|---|---|
| | IoU ↑ | Cham. ↓ | PSNR ↑ | SSIM ↑ | LPIPS ↓ | bunny | spot | lucy | arma. | dragon | woody | teapot |
| Backproj | 0.0598 | 0.0099 | 11.28 | 0.691 | 0.426 | 0.95 | 1.60 | 0.74 | 0.79 | 0.72 | 0.90 | 1.24 |
| RQ | 0.0135 | 0.0728 | 6.27 | 0.390 | 0.806 | 5.93 | 8.59 | 6.90 | 6.18 | 6.25 | 6.71 | 10.41 |
| TF-SS | 0.0177 | 0.0100 | 13.51 | 0.719 | 0.392 | 0.67 | 1.72 | 0.62 | 0.55 | 0.59 | 1.28 | 1.59 |
| TF-TS | 0.0483 | 0.0219 | 9.27 | 0.469 | 0.683 | 1.11 | 5.94 | 0.55 | 1.96 | 0.56 | 1.10 | 4.08 |
| **Ours** | **0.0908** | **0.0055** | **17.06** | **0.801** | **0.248** | **0.50** | **0.66** | **0.44** | **0.42** | **0.42** | **0.61** | **0.80** |

transceiver locations. For our model, we use twelve fully-connected layers, along with Müller et al. (2022) based hash encoding. We use 2200 rays and 170 depth samples for sampling. We use ADAM optimizer with learning rate $1e^{-5}$.

■ **Baselines:** We evaluate SpINR against four baselines varying in signal representation and physical modeling fidelity: (1) **TF-TS:** Time-domain forward model; loss computed directly in the time domain, (2) **TF-SS:** Time-domain forward model; loss computed in frequency domain after FFT, (3) **RQ:** Frequency-domain model using quantized range binning; computationally efficient but ignores complex phasor interactions, and (4) **Coherent Backprojection:** Voxel-based imaging via coherent summation of delayed contributions. All baselines share identical network architectures, encodings, sampling strategies, and training schedules.

■ **Metrics:** For evaluation, we use the standard metrics widely adopted in computer vision and inverse rendering. These include Intersection-over-Union (IoU), Chamfer Distance (CD), Hausdorff Distance (HD), Peak Signal-to-Noise Ratio (PSNR), Structural Similarity Index (SSIM), and Learned Perceptual Image Patch Similarity (LPIPS). For image-based metrics, we average over 20 random camera placements at a fixed radius. Together, they capture both geometric fidelity and perceptual image quality by comparing voxel grids, rendered views, and point cloud representations of reconstructed shapes.

■ **Results:** Fig. 2 presents a overall qualitative results of reconstruction with SpINR and compares with the baselines. Detailed results and analysis are as follows.

**(1) TF-TS vs. TF-SS:** We observe significant improvements in reconstruction metrics when using frequency-domain supervision (TF-SS) compared to purely time-domain supervision (TF-TS) both quantitatively (in Table 1) and qualitatively (in Fig. 2). Specifically, the frequency-domain loss leads to smoother convergence and more accurate geometric reconstructions as discussed earlier in the "Primer and Challenges" section.

**(2) Computation Efficiency:** SpINR not only improves the reconstruction quality but also significantly improves computational efficiency compared to the time-domain baseline. Specifically, our method scales better with increasing scene complexity, maintaining consistently lower runtime compared to the time-domain model, as shown in Fig. 3(a). For instance, at larger scene sizes, SpINR achieves approximately a 32% reduction in reconstruction time. As we reduce the number of synthesized bins to those overlapping the scene support, the runtime of SpINR decreases nearly linearly, confirming that the latency gains stem directly from frequency-selective spectral synthesis in our analytic forward model, rather than from implementation details alone.

**(3) Learning Stability:** Although both SpINR and TF-SS baseline ultimately compute loss in the frequency domain, we observe a significant performance gap in reconstruction accuracy and convergence speed. To investigate this, we analyze the gradient dynamics of the two approaches.

The importance of gradient flow in network design is well established. Deep architectures often suffer from vanishing or exploding gradients, motivating skip connections Balduzzi et al. (2017), normalized initialization Zhang et al. (2019), and sinusoidal activations for signal representation Sitzmann et al. (2020). These works demonstrate that preserving coherent and stable gradients across layers is crucial for trainability. Similar reasoning has been used to justify architectural choices in implicit neural representations Sitzmann et al. (2020), and sparse pruning strategies Wang et al. (2020). Following this motivation, we studied the computation graphs of both forward models. TF-

SS additional operations (time-domain simulation and FFT), creating fragmented gradient paths. In contrast, our frequency-domain forward model analytically predicts only relevant frequency bins, leading to a more direct gradient path and thus more stable gradient statistics, as shown in Fig. 3(b).

These observations mirror prior observations from studies on residual networks Balduzzi et al. (2017), gradient confusion Sankararaman et al. (2020), and signal representations Sitzmann et al. (2020), where architectural decisions that preserve or align gradients lead to better optimization and generalization. Our work extends this perspective to signal modeling for FMCW radar, showing that a forward model grounded in spectral domain physics not only improves interpretability but also facilitates learning through more stable and effective gradient flow.

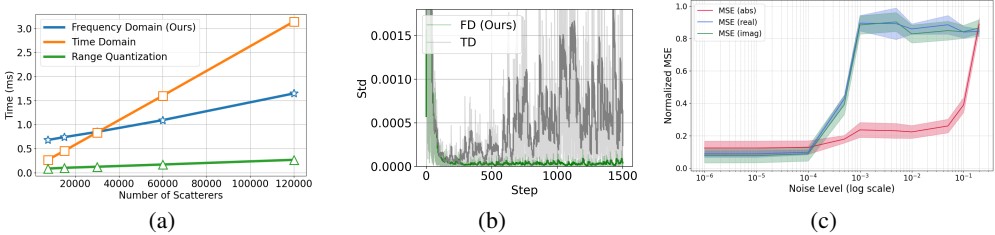

(a)  (b)  (c)

Figure 3: (a) Runtime vs. number of scatterers. (b) Gradient comparison for TF-SS and SpINR. (c) Effect of spatial perturbation on MSE metrics.

**(4) Scheduling of Losses:** Fig. 3 (c) analyzes how the MSE for different loss components—magnitude (abs), real, and imaginary—responds to increasing spatial noise. We observe that magnitude loss remains stable and discriminative even under large perturbations (up to 10cm), making it ideal for guiding coarse geometry early in training. In contrast, real and imaginary components become sensitive only at finer scales (below 1mm), capturing high-frequency phase details.

Based on this, we adopt a staged loss strategy: we begin training with magnitude-only loss to achieve coarse alignment, and gradually introduce the real and imaginary terms to refine fine-grained structure. This approach leads to faster convergence and improved reconstruction fidelity across varying noise levels.

**(5) SpINR vs. Learning Based Baselines:** We benchmark our method against two closest learning-based approaches despite their distinct target applications – NVR Reed et al. (2023) and RadarHD Prabhakara et al. (2022). Although these methods differ slightly in their objectives—NVR targets volumetric rendering from acoustic signals, and RadarHD aims to enhance long-range mmWave radar point clouds—they represent the nearest available learning-based baselines. Compared to NVR, SpINR achieves a 52.6% reduction in Chamfer Distance, highlighting superior volumetric reconstruction fidelity. RadarHD, in contrast, did not converge to meaningful 3D reconstructions in our experimental setup as shown in Fig. 4 (a). Some past works like Rafidashti et al. (2025), DART Huang et al. (2024), and Radar Fields Borts et al. (2024b) operate on post-processed 2D range–azimuth tensors from multi-Tx/Rx MIMO stacks, whereas our focus is 1D raw chirps with cylindrical apertures and thus out of scope of our comparison.

### 5.1 ABLATION STUDY

**(1) Smoothness and Sparsity Regularization:** Figure 4 (b) shows the impact of smoothness and sparsity regularization on reconstruction quality. Without regularization, reconstructions contain noisy artifacts, fragmented surfaces, and faint shell-like structures around the true geometry. In contrast, adding smoothness and sparsity regularization effectively removes these artifacts, producing cleaner surfaces and sharper object boundaries. Unless otherwise stated, we use the same smoothness and sparsity weights $(\beta, \gamma)$ for all scenes and radar configurations; we do not retune them per experiment. Regularization significantly enhances geometric fidelity by enforcing spatial coherence and compactness, demonstrating its necessity in high-frequency radar reconstruction scenarios.

**(2) Sampling Strategies:** Fig. 5 (a) shows the reconstructions for (from left to right) ray-based uniform sampling, ray-based cubic sampling, and random sampling. Both ray-based cubic sampling and random sampling, where we aim to maintain uniform sampling density perform better. Additionally, random sampling can achieve equivalent sampling density as ray-based methods with only 10.7% sampled points compared to ray-based approaches.

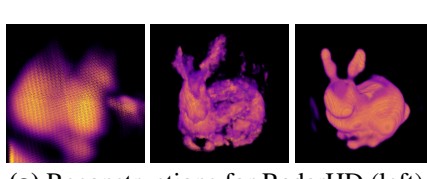

**(a)** Reconstructions for RadarHD (left), NVR (center), and SpINR (right).

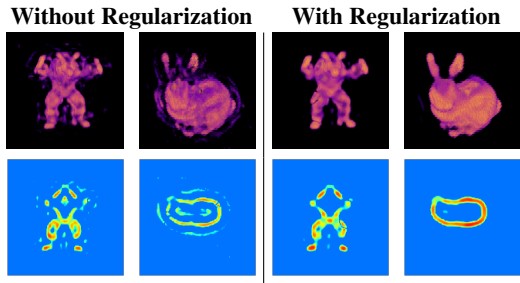

**(b)** Effect of regularization. Top row - 3D reconstructions, bottom row - XZ slice plots of the reconstruction.

Figure 4: (a)Baseline comparison (b) Effect of regularization

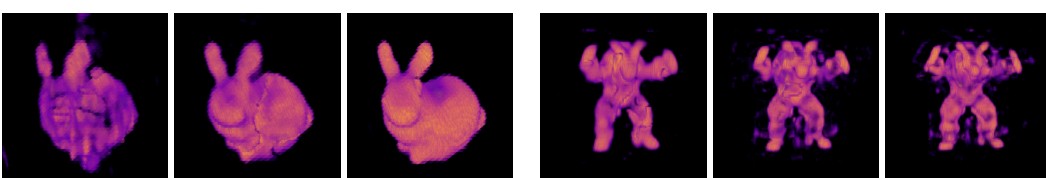

(a) Effect of Uniform ray-based (left), Cubic ray-based (center), and Random (right).

(b) Effect of start frequencies at 2, 3, and 4 GHz.

Figure 5: sampling strategies (left) and start-frequency effects (right).

**(3) Start Frequencies:** Fig. 5 (b) and Tab 2 illustrate the impact of start frequency $f_0$ on reconstruction quality. While bandwidth governs range resolution, higher start frequencies introduce greater phase wrapping due to reduced wavelength, making sub-bin positions harder to distinguish. This manifests as a multiplicative term $e^{i\phi}$, where $\phi = 2\pi f_0 \tau$. As seen in the figure, reconstructions at 2 GHz are clean and well-defined, but increasing $f_0$ to 3 GHz and 4 GHz results in growing artifacts and ghost-like duplications, consistent with phase aliasing effects. These results also motivate the need for regularization techniques to suppress sub-bin ambiguities.

| Frequencies | IoU (↑) | Chamfer (↓) | Hausdorff (↓) | PSNR (↑) | SSIM (↑) | LPIPS (↓) |
|---|---|---|---|---|---|---|
| 0 | 0.1312 | 0.0050 | 0.0507 | 16.6148 | 0.6971 | 0.2799 |
| 100 | 0.1311 | 0.0050 | 0.0507 | 16.6203 | 0.6974 | 0.2793 |
| 1e6 | 0.1303 | 0.0050 | 0.0507 | 16.5800 | 0.6950 | 0.2849 |
| 100e6 | 0.1328 | 0.0049 | 0.0507 | 16.7841 | 0.6960 | 0.2825 |
| 500e6 | 0.1243 | 0.0054 | 0.1179 | 14.8324 | 0.5537 | 0.5336 |
| 1e9 | 0.0313 | 0.0278 | 0.1514 | 7.2550 | 0.3479 | 0.7283 |

Table 2: Effect of Start frequency on Reconstruction without regularization

## 6 CONCLUSION

In this paper, we present SpINR, a novel framework for volumetric reconstruction using FMCW radar data. By integrating a fully differentiable forward model operating in the frequency domain with implicit neural representations (INRs), SpINR effectively leverages the inherent linear relationship between beat frequency and scatterer distance. Our formulation is specifically tailored to the phase-critical regime of small, near-field tabletop volumes, where wavelength is comparable to geometric detail and complex spectral information is essential; extending these ideas to large-scale scenes, other sensing geometries and real-world data is an interesting and important direction for future work. This approach not only facilitates more efficient and accurate learning of scene geometry but also enhances computational efficiency by focusing on relevant frequency bins.

## 7 REPRODUCIBILITY

Anonymized code and demo datasets will be available on our webpage (`https://anon61823-star.github.io`). We provide details about comparison with other algorithms to facilitate the reproducing of our results. All details about the hyperparameters, environment specifications, and real-world experiment setup are provided in the appendix or the website.

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

## A    DISCLOSURE OF LLM USAGE FOR WRITING

Large Language Models (LLMs) were used solely for grammar refinement and polishing of the manuscript text. All ideas, technical content, experimental design, and analysis were independently developed by the authors without LLM assistance.

## B    DERIVATION OF SPECTRAL TRANSFORMATION FOR FORWARD MODEL

To clarify the mathematical foundation of our *spectral forward model* in SpINRv2 (Ours), we revisit the discrete Fourier analysis of a single complex tone. This explicit derivation highlights two properties that are crucial for radar reconstruction: (i) how a finite-length DFT distributes energy into neighbouring bins when a scatterer's beat frequency is *off-bin*, and (ii) why modelling both magnitude *and* phase is essential for physically faithful supervision.

**Signal model.**    We consider the sampled beat signal from one ideal point scatterer,

$$S_t \; = \; M \, e^{\mathrm{i}(\alpha t + \phi)}, \qquad \alpha \; = \; \frac{2\pi f}{N}, \quad t = 0, \ldots, N-1, \tag{3}$$

where $M$ is the amplitude, $\phi$ the initial phase, $f$ the (normalised) frequency and $N$ the analysis window length.

**DFT definition.**    The unit-normalised DFT coefficient at bin index $k$ is

$$Z_k \; = \; \frac{1}{N} \sum_{t=0}^{N-1} S_t \, e^{-\mathrm{i}\beta_k t}, \qquad \beta_k \; = \; \frac{2\pi k}{N}. \tag{4}$$

**Forming a geometric series.**    Substituting $S_t$ and collecting constants gives

$$Z_k \; = \; \frac{M}{N} \, e^{\mathrm{i}\phi} \sum_{t=0}^{N-1} \bigl(e^{\mathrm{i}(\alpha - \beta_k)}\bigr)^t, \tag{5}$$

which is a finite geometric series with common ratio $r = e^{\mathrm{i}(\alpha - \beta_k)}$.

**Closed-form sum.**    Using the standard sum formula $\sum_{t=0}^{N-1} r^t = (1 - r^N)/(1 - r)$ we obtain

$$Z_k \; = \; \frac{M}{N} \, e^{\mathrm{i}\phi} \, \frac{1 - e^{\mathrm{i}(\alpha - \beta_k)N}}{1 - e^{\mathrm{i}(\alpha - \beta_k)}}. \tag{6}$$

**Eliminating the bin phase.**    Because $\beta_k N = 2\pi k$ is an integer multiple of $2\pi$, $e^{-\mathrm{i}\beta_k N} = 1$ and equation 6 simplifies to the closed form used in the main paper:

$$Z_k \; = \; \boxed{\frac{M}{N} \, e^{\mathrm{i}\phi} \, \frac{1 - e^{\mathrm{i}\alpha N}}{1 - e^{\mathrm{i}(\alpha - \beta_k)}}}. \tag{2 revisited}$$

**Interpretation.** If the scatterer frequency *exactly* matches a bin centre ($\alpha = \beta_k$), both numerator and denominator of equation 2 revisited vanish; evaluating the limit (or returning to the original sum) yields $Z_k = M\,e^{i\phi}$ for that bin and $Z_{k'} = 0$ elsewhere. For all *off-bin* frequencies ($\alpha \neq \beta_k$), equation 2 revisited predicts a sinc-shaped spectral envelope—the familiar *spectral leakage* modelled in SpINR. This can also be seen in Fig. 6 and 7. Capturing this leakage, including complex phase rotations, is what allows our frequency-domain supervision to remain physically consistent even when a scatterer's true delay does not align with a DFT bin centre.

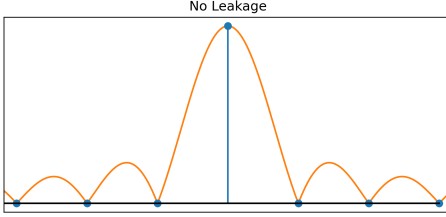

Figure 6: When the $\alpha = \beta_k$, there is no leakage

Figure 7: When the $\alpha \neq \beta_k$, there is leakage and in neighbouring bins

## C  ADDITIONAL EVALUATION

### C.1  ADDITIONAL QUALITATIVE RESULTS FOR RECONSTRUCTION

In this section we show some additional results for scene reconstruction with SpINRv2 (Ours) and othe baselines.

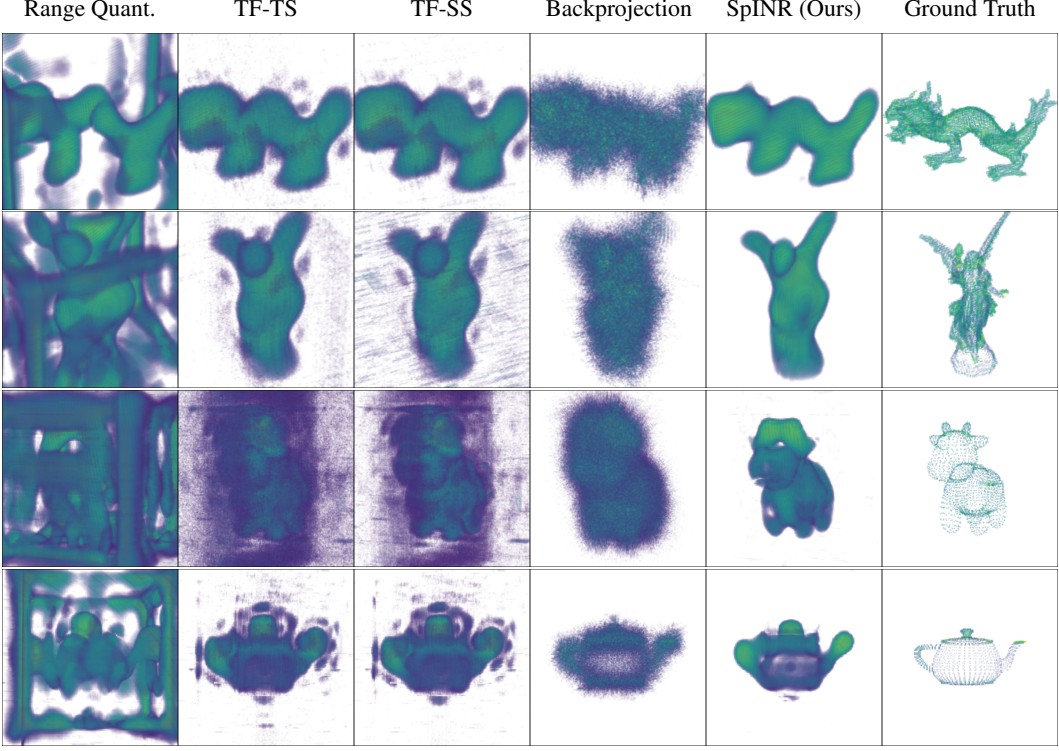

Figure 8: Comparison of volumetric reconstructions

## C.2 COMPUTATION EFFICIENCY OF SPINRV2 (OURS) VS. TF-TS VS. RQ

Fig. 9 shows us a comparison between computation efficiency and quality of reconstruction for various models. SpINRv2 (Ours) outperforms Time domain forward in both runtime efficiency and chamfer distance. Range Quantization is much faster than SpINRv2 (Ours)'s forward model but severely underperforms in reconstruction quality.

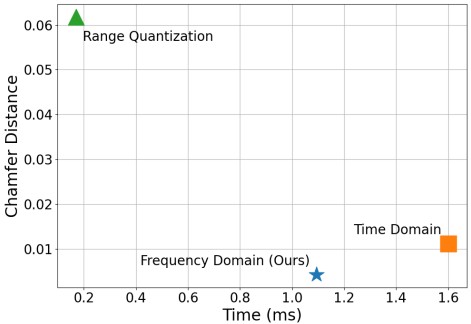

Figure 9: We compare the three forward models for their runtime and volume reconstruction accuracy. Even though the Range Quantization has the fastest runtime, it severe underperforms in volume reconstruction as signified by the chamfer distance (lower better).

## C.3 RECONSTRUCTION QUALITY OF SPINRV2 (OURS) VS. CLASSICAL COHERENT BACKPROJECTION

To further validate the effectiveness of our proposed frequency-domain forward model, we compare it against a classical reconstruction baseline: coherent backprojection (CBP). This method discretizes the scene into voxels and accumulates backprojected energy from all Tx paths by aligning phases based on path delays. While physically grounded, CBP has significant limitations that our learning-based approach addresses.

(1) Inherent Discretization and Aliasing: CBP relies on uniform voxel grids and discrete binning of round-trip delays, which often leads to aliasing artifacts and blurring, especially when the sampling aperture is non-uniform or sparse. Our method, by contrast, uses a continuous volumetric representation (via neural fields) and avoids explicit voxelization, enabling sub-voxel accuracy in both representation and rendering.

(2) Lack of Differentiability and Learning: CBP is a purely geometric method—it does not learn from data or optimize a forward model. As a result, it is highly sensitive to aperture coverage, noise, and partial views. Our method incorporates a fully differentiable, physics-informed forward model, enabling gradient-based optimization that refines the scene representation to best match all measurements.

(3) Better Use of Redundant Views: In cases with dense or overlapping Tx-Rx observations, CBP merely accumulates energy, often leading to oversmoothened reconstructions. In contrast, our method learns to fuse redundant observations through a shared implicit scene representation, enabling better consistency across viewpoints and preservation of fine geometry.

As shown in our quantitative results (Tables 1,and 2, SpINR achieves significantly better reconstruction compared to CBP.

In summary, while CBP serves as a useful classical baseline, it lacks the flexibility, expressiveness, and robustness offered by our differentiable frequency-domain formulation paired with neural reconstruction. This synergy between physics-based modeling and data-driven learning is key to achieving high-fidelity 3D volumetric reconstructions from FMCW radar measurements.

## C.4 EFFECT OF BANDWIDTH CHANGE ON RECONSTRUCTION QUALITY

The spatial resolution of FMCW radar systems is fundamentally limited by the chirp bandwidth $B$, with the minimum distinguishable range given by $\frac{c}{2B}$. As the bandwidth decreases, this resolution limit coarsens, which traditionally leads to severe blurring in reconstruction techniques such as backprojection. However, SpINR is not bound by explicit voxel grids or range quantization. Instead, it learns a continuous implicit representation of the scatterer field. This allows the network to infer and preserve spatial structure even under low-bandwidth conditions, where conventional methods lose detail.

Figure 10 compares reconstructions from SpINR and classical backprojection as the bandwidth is reduced from 4 GHz down to 40 MHz—a 100x decrease. While both methods suffer some degradation in spatial sharpness, SpINR exhibits a notably graceful degradation. Key object features remain preserved at lower bandwidths, and the reconstructions retain their overall shape and topology. In contrast, backprojection rapidly collapses into severely blurred and aliased outputs. These results highlight the robustness of the learned representation in SpINR to physical resolution limits imposed by the sensor, demonstrating its utility in bandwidth-constrained regimes.

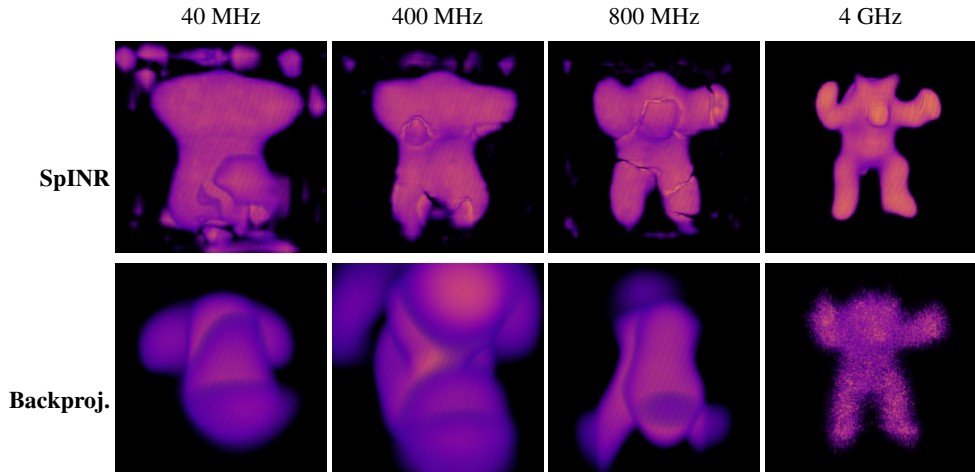

Figure 10: Comparison of volumetric reconstructions for different bandwidths (4GHz, 800 MHz, 400 MHz, 40 MHz) using SpINR and classical backprojection. SpINR shows a graceful degradation as compared with backprojection.

**Effect of Start Frequency on Reconstruction.** Table 2 highlights how the choice of start frequency $f_0$ influences reconstruction quality in the absence of regularization. Although range resolution is determined by the signal bandwidth, increasing $f_0$ leads to shorter wavelengths and thus more severe phase wrapping—making it difficult to resolve scatterers at sub-bin offsets. This effect appears as a multiplicative phase term $e^{i\phi}$, where $\phi = 2\pi f_0 \tau$. As shown in the Fig. 11, as the frequencies increase, the loss landscape becomes non-flat and the ambiguity increases. These observations further motivate the use of regularization to mitigate ambiguities arising from off-bin scatterer positions.

**Encoder ablation.** To assess whether our choice of multi-resolution hash encoding is critical, we perform an encoder ablation on the bunny scene, varying both the hash configuration and the underlying coordinate encoding. Our *Baseline Hash encoding* uses the same multi-resolution hash grid as in the main experiments (Instant-NGP style, with multiple levels and a large hash table). In *levels8* and *levels20* we decrease or increase the number of hash levels while keeping other hyperparameters fixed, and in *smallhash20* we reduce the hash table size. In *Hash scale1p2* we increase the base resolution (hash scale) by a factor of 1.2. As non-hash alternatives, *Freq6* replaces the hash grid with low-frequency Fourier features using 6 frequency bands, *Identity* uses raw 3D coordinates (normalized to the scene bounds) without any encoding, and *Triangle6* uses a simple triangular-

wave positional encoding with 6 harmonics. Table 3 reports correlation, slice-wise SSIM, Chamfer distance, Hausdorff distance, IoU, and Dice score for these variants. All hash-based configurations achieve very similar performance to the baseline, indicating that a standard hash-encoding INR has sufficient capacity and is robust to moderate changes in its hyperparameters. In contrast, the non-hash encodings (Freq6, Identity, Triangle6) collapse completely, with near-zero Corr/SSIM and extremely poor geometric metrics. This supports our design choice: the multi-resolution hash encoding is an effective and relatively generic representation for the volumetric reflectivity field, and the main gains of SpINR come from the analytic spectral forward model and training scheme rather than from a specialized coordinate encoding.

| Experiment | Corr ($\uparrow$) | SSIM (slice) ($\uparrow$) | Chamfer ($\downarrow$) | Hausdorff ($\downarrow$) | IoU ($\uparrow$) | Dice ($\uparrow$) |
|---|---|---|---|---|---|---|
| Base Hash encoding | 0.396 | 0.957 | 1.943 | 6.708 | 0.124 | 0.221 |
| Freq6 | 0.015 | 0.001 | 9.253 | 20.890 | 0.003 | 0.007 |
| levels8 | 0.391 | 0.950 | 1.727 | 6.212 | 0.119 | 0.213 |
| levels20 | 0.397 | 0.951 | 1.760 | 6.243 | 0.124 | 0.221 |
| Hash scale1p2 | 0.401 | 0.930 | 1.490 | 5.987 | 0.123 | 0.219 |
| smallhash20 | 0.396 | 0.948 | 1.743 | 6.322 | 0.124 | 0.220 |
| Identity | -0.071 | 0.011 | 22.915 | 21.491 | 0.001 | 0.002 |
| Triangle6 | 0.010 | 0.000 | 8.104 | 19.811 | 0.003 | 0.006 |

Table 3: Encoder ablation on the bunny scene. Hash-based configurations remain close to the baseline across metrics, while simple non-hash encodings (Freq6, Identity, Triangle6) fail catastrophically.

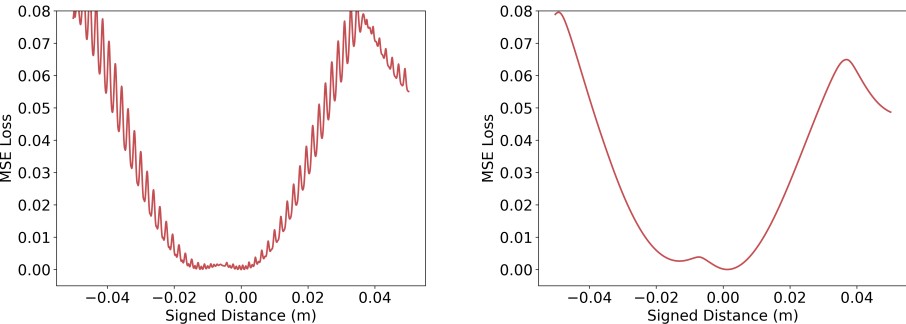

Figure 11: Loss when $\lambda/4 < \Delta r$ vs. loss when $\lambda/4 > \Delta r$. As the start frequency increases the loss landscaped becomes non-flat due to ambiguity.

## C.5 EVALUATION ON A HIGH-FREQUENCY SHEET BENCHMARK

To evaluate the spatial resolution limits of SpINR, we introduce a synthetic benchmark object designed to gradually vary in surface detail. Specifically, we construct a vertical sheet where sinusoidal undulations increase in frequency along the y-axis (Fig. 12). The initial part of the object features smooth, low-frequency ripples, while the latter part contains increasingly fine patterns that challenge the system's resolving capacity.

We render volumetric reconstructions using SpINR and compare the learned scatterer field with the ground-truth surface. As shown in Fig. 13, SpINR accurately captures coarse structures in the first half, with clear alignment between the predicted maxima and the ground-truth scatterer locations. However, in the high-frequency regions, the system begins to miss fine oscillations, resulting in a loss of structural fidelity and localized blurring.

## C.6 ADDITIONAL RESULTS ON GRADIENT ANALYSIS

As explained in the main paper, We visualize the gradient statistics of both forward models (Figure 15). Our method (green) exhibits significantly more stable gradient statistics than the time-

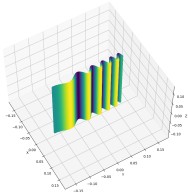 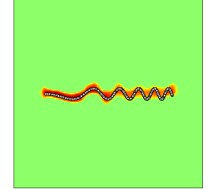 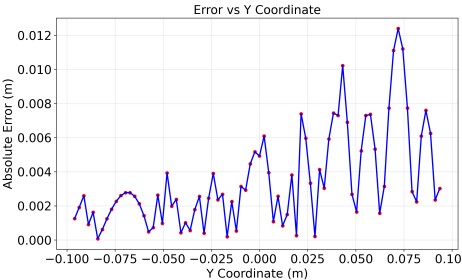

Figure 12: (a) Ground-truth 3D sheet geometry

Figure 13: (b) Overlay of predicted maxima (red) with GT (white)

Figure 14: (c) Error vs. vertical Y-axis

domain model (gray). The variance remains controlled throughout training, avoiding sudden spikes or vanishing behavior.

Our empirical observations parallel a broader body of work showing that architectural choices which preserve or align gradients promote both stable optimization and stronger generalization. In the context of FMCW radar signal modeling, we extend this principle by introducing a forward model rooted in spectral-domain physics. The resulting design not only enhances interpretability but also yields smoother, more effective gradient flow, thereby facilitating faster and more reliable training.

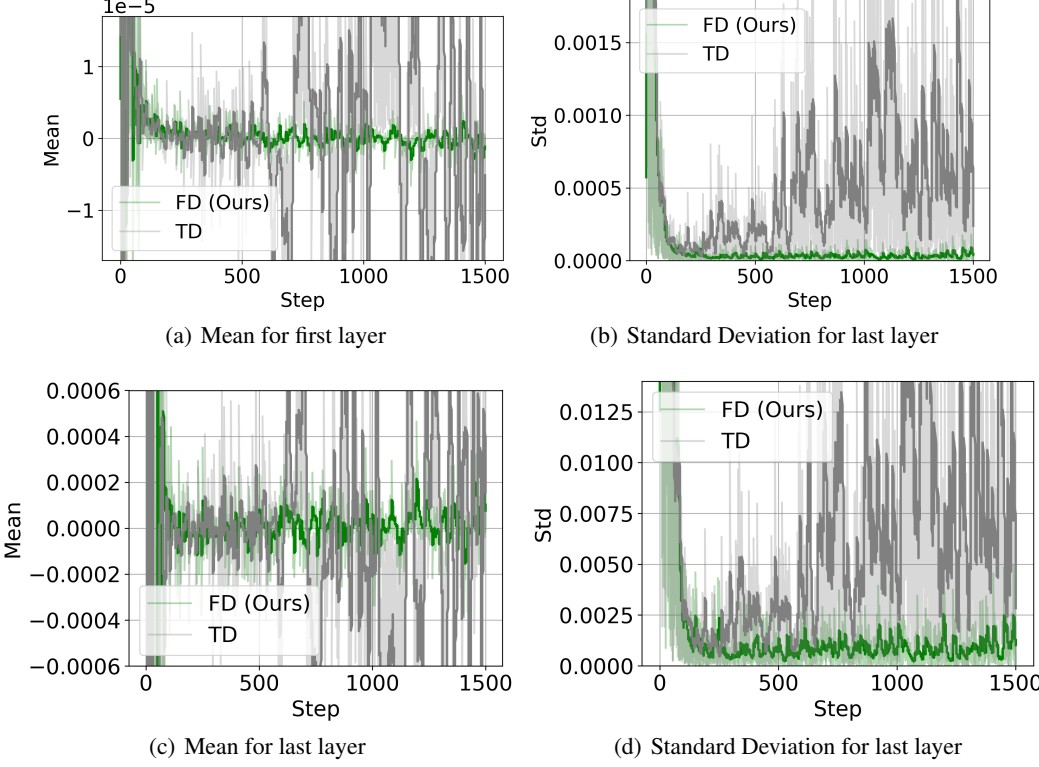

Figure 15: We log the (a) mean and (b) standard deviation of the gradients for the first layer w.r.t. the number of training steps. The mean and standard deviation are more stable for our proposed method. For the time domain forward model the the gradient tends to explode. We observe a similar trend for all the subsequent layers.

| Parameter | Default Value | Description |
|---|---|---|
| **Learning & Optimization** | | |
| learning_rate | 1e-5 | Learning rate for network training |
| num_epochs | 345600 | Number of transmitter locations to train on |
| accum_grad | 1 | Gradient accumulation steps |
| batch_size | 1 | Training batch size |
| **Network Architecture** | | |
| num_layers | 12 | Number of hidden layers in the network |
| num_neurons | 256 | Number of neurons per hidden layer |
| **Loss Weights** | | |
| scale_factor | 0.00005 | Overall loss scaling factor |
| mse_weight | 0.01 | Weight for MSE loss component |
| smooth_loss | 1e4 | Weight for smoothness regularization |
| sparsity | 5e3 | Weight for sparsity regularization |
| **Occlusion & Geometry** | | |
| occlusion_scale | 1e0 | Scale factor for occlusion handling |
| beamwidth | 90 | Radar beam width in degrees |
| **Sampling & Rays** | | |
| num_scatterers | 40000 | Number of scatterer points sampled per iteration for random sampling |
| num_rays | 2200 | Number of rays processed per batch |
| max_voxels | 15000 | Maximum number of voxels to process |
| sampling_distribution_uniformity | 1.0 | Uniformity of sampling distribution |

Table 4: Default hyperparameters used in training and evaluation.

## D   REPRODUCIBILITY

The code for dataset creation, model training and evaluation can be found in the "MMwave_recon" folder. The folder has a "README.md" which contains all the steps for running the code.

## E   HYPERPARAMETERS

