# OpenReview forum: "Spectral Domain Neural Reconstruction for Passband FMCW Radars"
_ICLR.cc/2026/Conference — Submitted to ICLR 2026_

### Official Review · Reviewer_UmeR · 2025-10-30

**Soundness:** 2
**Presentation:** 1
**Contribution:** 1
**Rating:** 2
**Confidence:** 4

**Summary:**

The paper presents a method for 3D volumetric reconstruction from FMCW radar data.
The approach models the radar signal in the frequency domain using a DFT-based formulation, and employs a neural network that predicts scattering intensity by being trained on the complex frequency response, with regularization terms for smoothness and sparsity. The main claim for novelty and performance gain stem from modeling radar physics directly in the spectral domain, rather than relying on time-domain representations.

**Strengths:**

1. The paper’s application of volumetric rendering to radar data is interesting.
2. The inclusion of smoothing and sparsity regularization to address radar spectral issues is an effective way to enhance the method’s stability and robustness.

**Weaknesses:**

The core novelty claimed by the paper lies in transforming the FMCW signal to the frequency domain and modeling the reflectivity physics in the spectral domain. However, this transformation is a standard step in FMCW processing, where the beat signal is typically converted to the spectral domain to achieve signal compaction and improved signal-to-noise ratio. Since this procedure is common practice, and the rendering process itself is also not new, I find the overall level of novelty limited, both in general and specifically from a machine learning perspective.

**Questions:**

Given that spectral-domain processing is standard for FMCW signals, what is the core novelty of the paper?

---

> ### Author Response · Authors · 2025-11-22
>
> # Summary of Changes and Clarifications
>
> We thank reviewer UmeR for appreciating our contributions. We respectfully but strongly disagree with the novelty assessment, which we believe reflects a misinterpretation of SpINR.
>
> ### 1. “Spectral-domain processing is standard” vs. what SpINR actually does
>
> We agree FFT/DFT are differentiable and FMCW pipelines routinely transform beat signals to the spectral domain, but the critique conflates:
>
> 1. **Standard FMCW processing**, which
>    – mixes, windows, and FFTs to obtain range(-Doppler) profiles;
>    – typically uses **real-valued range–Doppler / range–azimuth heatmaps** as network inputs;
>    – treats spectral leakage and off-bin behavior as artifacts of the hardware/FFT rather than as part of an explicit forward model.
>
> 2. **SpINR’s analytic spectral forward model**, which
>    – does **not** treat pre-FFT time samples as the native space, but defines a direct mapping
>      \(\sigma(\mathbf{x}) \longrightarrow Z_k(\sigma)\),
>      where each DFT bin is an integral over the scene with a kernel encoding
>      • FMCW modulation,
>      • finite-time window \(\Rightarrow\) **Dirichlet-kernel leakage**,
>      • passband phase from round-trip delay and chirp slope;
>    – implements this as a **closed-form differentiable layer** from the 3D field to complex spectra, without explicitly synthesizing the full beat signal;
>    – is parameterized by bin index \(k\), so we **evaluate only bins** whose ranges intersect the tabletop volume (e.g., first 16–32 of 256), yielding the latency reduction in Fig. 3;
>    – trains on **complex spectra** with a staged magnitude\(\rightarrow\)complex loss that exploits different stability regimes of magnitude and phase.
>
> We do **not** claim that “using FFT” or “working in frequency” is novel. The contribution is **formulating the FMCW DFT as a physics-accurate, differentiable rendering operator for a neural volumetric field, with complex-bin supervision and frequency-selective synthesis in the small near-field regime.**
>
> ---
>
> ### 2. Novelty from a ML perspective
>
> We disagree that the ML novelty is limited. Rather than a new backbone, we provide **physics-informed design of the forward operator, losses, sampling, and regularization** that materially changes optimization:
>
> 1. **Physics-informed loss staging (magnitude \(\rightarrow\) magnitude+phase).**
>    Sensitivity analysis (Fig. 3(c)) shows
>    – magnitude MSE is relatively stable at coarse spatial scales,
>    – phase MSE is extremely sensitive at sub-mm scales.
>    This motivates staging: first optimize magnitude for coarse geometry, then add complex loss for fine structure, stabilizing training at high start frequencies.
>
> 2. **Improved gradient flow via the spectral forward model.**
>    Gradient-flow analysis (Fig. 3(b), App. C.6) shows
>    – time-domain synthesis + FFT yields fragmented, noisy gradients and slow convergence,
>    – the analytic spectral model yields more coherent gradients aligned with geometry.
>    The forward model thus acts as a better parameterization, **smoothing the optimization landscape** and improving convergence.
>
> 3. **Geometry-aware sampling for cylindrical apertures.**
>    We introduce the cubic radial law
>    \(r(u) = \big(u(r_{\text{far}}^3 - r_{\text{near}}^3) + r_{\text{near}}^3\big)^{1/3}\)
>    to equalize point density across depth for cylindrical synthetic apertures, avoiding under-sampling near the center and over-sampling near the far boundary. This improves coverage and reconstructions—a concrete ML contribution tailored to this geometry.
>
> 4. **Radar-specific regularization for sub-bin ambiguity.**
>    The smoothness and sparsity terms are not generic “L2 + L1” penalties. They are chosen to
>    – regularize phase-wrapped, off-bin responses via spatial continuity,
>    – encode that most of the tabletop volume is empty.
>    Fig. 4(b) shows that removing them produces ghost scatterers and shell artifacts, whereas including them yields clean volumes.
>
> Together, these elements show that the contribution is not “do a standard FFT and train an INR,” but **use FMCW physics to design a forward operator and training scheme that enable stable, high-fidelity neural volumetric reconstruction where naive time-domain or magnitude-only approaches fail.**
>
> ---
>
> ### 3. Regime and scope
>
> Our target regime is **small, near-field tabletop volumes** at multi-GHz carrier frequencies, where
>
> - wavelength is comparable to geometric detail,
> - baseband range-bin spacing is coarse relative to the structures of interest,
> - off-bin Dirichlet leakage and phase variation across bins are crucial for fine geometry.

---

> > ### Author Response · Authors · 2025-11-22
> >
> > # Answer to explicit questions
> >
> > ## Q1.  “Given that spectral-domain processing is standard for FMCW signals, what is the core novelty of the paper?”
> >
> > A1. The novelty of SpINR is **not** “transforming the signal to the frequency domain,” which is indeed standard in FMCW processing. The core contribution is that we:
> >
> > 1. **Derive and implement a closed-form, differentiable, complex-valued FMCW DFT forward model** that maps a continuous 3D reflectivity field (\sigma(\mathbf{x})) directly to discrete complex DFT bins (Z_k), explicitly modeling **Dirichlet-kernel spectral leakage** and passband phase as functions of scatterer position and chirp parameters;
> > 2. **Supervise an implicit neural volumetric field directly on these complex spectral bins**, with a staged magnitude→complex loss tailored to the sensitivity of phase in the small, near-field regime;
> > 3. **Exploit frequency-selective synthesis**: because the forward model is parameterized directly by frequency index (k), we synthesize only those bins whose range support overlaps the bounded tabletop volume, which yields the measured latency gains in Fig. 3 and is not available to time-domain synthesis + FFT baselines;
> > 4. Apply this framework to the **phase-critical regime of small, near-field tabletop volumes**, where wavelength is comparable to geometric detail and off-bin complex structure is essential for high-fidelity reconstruction.
> >
> > This is substantially different from the “standard” FMCW use of FFT as a pre-processing step.
> >
> > In summary, our contribution lies in the **analytic FMCW DFT layer with Dirichlet leakage, frequency-selective synthesis, complex spectral supervision, small-scale scene synthesis, and physics-informed ML design** tailored to the small, near-field volumetric reconstruction problem. Most prior neural radar works focus on **large automotive / indoor scenes with planar MIMO arrays**, using real-valued range–azimuth / range–Doppler maps. Those works and SpINR are **complementary**: they tackle coarse-grained large-scale perception, whereas SpINR is designed for the **phase-critical small-volume regime**, where a physics-accurate spectral forward model with complex supervision is necessary.
> >
> > We believe this constitutes substantial novelty and addresses the reviewer’s concern on both signal-processing and machine-learning grounds.

---

> > > ### Author Response · Authors · 2025-11-25
> > > **Follow-up on author response**
> > >
> > > Dear Reviewer UmeR,
> > > Thanks again for your feedback regarding our paper. We have responded to your initial comments/questions and have incorporated them accordingly into our revised manuscript.
> > >
> > > We would greatly appreciate you taking a moment to review our response, reassess our revised manuscript from all aspects, share your thoughts, and update your score accordingly. We sincerely believe that our paper has been strengthened a lot thanks to the feedback. We look forward to hearing from you.

---

### Official Review · Reviewer_7oUU · 2025-10-31

**Soundness:** 2
**Presentation:** 1
**Contribution:** 2
**Rating:** 4
**Confidence:** 4

**Summary:**

This paper proposes a volumetric reconstruction framework for FMCW radar using a neural network with a forward model defined in the frequency domain via the Discrete Fourier Transform (DFT). Since FMCW radar image inherently lies in the frequency domain, defining the loss in that domain is reasonable.

**Strengths:**

1. The paper proposes a volumetric reconstruction framework for FMCW radar using a neural network with a forward model defined in the frequency domain via the DFT.

2. Since FMCW radar data inherently lies in the frequency domain, defining the loss in that domain is reasonable and conceptually consistent.

3. The overall idea is straightforward and easy to follow.

**Weaknesses:**

1.	Paper structure and clarity:
The manuscript is not well structured. A large portion of the content focuses on FMCW vs. pulse radars, which is not directly relevant to the core contribution.
The discussion on beat-signal challenges is repetitive and does not provide new insight.
There are noticeable typos throughout the manuscript, which negatively impact readability.
2.	Overemphasis on well-known theory:
The DFT formulation and explanation of spectral leakage are standard knowledge and could be shortened significantly.
Starting directly from the section “Forward Model with Spectral Synthesis” would already give sufficient context.
3.	Lack of meaningful comparisons:
The paper does not compare against sparse-reconstruction baselines or state-of-the-art deep learning methods for inverse problem.
TF-TS, TF-SS, and BP alone are insufficient to demonstrate the advantages of the proposed approach.
NVR and RadarHD are not suitable comparisons since they address different tasks and use different signal models.

4.	No real-world validation:
All experiments are simulation-based.
Without real data, it is difficult to assess practical effectiveness, robustness, or applicability in real FMCW systems.

5.	Missing In-depth discussion:
There is no analysis of how start frequency ( f0 ) and other radar configuration impact reconstruction quality.
The influence of DFT size (Dirichlet kernel width) on model performance is not explored.
These analyses could provide deeper insight and potentially lead to better design choices.

**Questions:**

1.	How robust is the proposed model to different radar configurations (e.g., start frequency f0, bandwidth B, chirp rate, and DFT size/Dirichlet kernel width)? If the model is trained on one configuration, can it generalize to others, or would re-training be required?
2.	Will the method work with real FMCW radar measurements, and do the authors plan to validate on hardware?
3.	Are there reasons sparse-reconstruction or modern neural inverse methods were not included in the comparisons?

---

> ### Author Response · Authors · 2025-11-22
>
> ## **Summary of Changes and Clarifications**
>
> We thank Reviewer 7oUU for highlighting that our core idea of frequency-domain formulation for FMCW radar. However, we respectfully disagree with several characterizations in the review and believe there may be fundamental misunderstandings about our contributions and problem setting. We address each concern in detail below.
>
> ---
>
> ## W.1 Structure, primer, and typos
>
> > “The manuscript is not well structured… A large portion of the content focuses on FMCW vs. pulse radars… DFT formulation and explanation of spectral leakage are standard knowledge and could be shortened significantly.”
> >
>
> Our intended audience includes ML researchers who may not be completely familiar with FMCW radar or the distinction between pulse-echo time-of-flight vs. chirp-based systems. The comparison between pulse and FMCW in Sec. 3 is not meant as “background for its own sake,” but to motivate why **existing neural volumetric frameworks designed for pulse-based SONAR/SAS (e.g., NVR)** cannot be directly ported to FMCW beat signals, and why time-domain supervision is ill-conditioned in our setting.
>
> That said, we agree that some of this material can be made more concise and moved to the appendix without weakening the main argument. In the revision we have:
>
> - Compressed the FMCW vs. pulse primer;
> - Start Sec. 4 (“Forward Model with Spectral Synthesis”) earlier and make clear up front that our main object is the **analytic FMCW DFT forward operator** from (\sigma(\mathbf{x})) to complex (Z_k);
> - Perform a thorough proofreading pass to remove typos and streamline wording.
>
> We note that the other reviewers did not flag serious structural issues (Reviewer tv6w explicitly called the exposition “smooth” and Reviewer x9mD rated presentation as “good”), but we agree that tightening Sec. 3 and the DFT discussion will improve clarity.
>
> ---
>
> ## W.2 “Overemphasis on well-known theory” vs. analytic forward model
>
> > “The DFT formulation and explanation of spectral leakage are standard knowledge… Starting directly from ‘Forward Model with Spectral Synthesis’ would already give sufficient context.”
> >
>
> We fully agree that DFT and spectral leakage are standard signal-processing concepts. But our contribution is - ”**how to use it as a differentiable forward operator in a neural volumetric reconstruction framework for FMCW”.**
>
> Standard FMCW practice is:
>
> - Mix, window, apply FFT to obtain range(-Doppler) profiles;
> - Often use **real-valued range–Doppler / range–azimuth heatmaps** as inputs to downstream networks;
> - Treat spectral leakage as a nuisance of the hardware/FFT rather than as a modeled phenomenon.
>
> In contrast, SpINR:
>
> 1. **Derives each complex DFT bin (Z_k) analytically and independently** as an integral over the 3D reflectivity field (\sigma(\mathbf{x})), with a kernel that encodes:
>     - FMCW modulation,
>     - finite time support → **Dirichlet kernel leakage**,
>     - passband phase as a function of round-trip delay and chirp slope.
> 2. **Implements this mapping as a closed-form differentiable layer** from (\sigma(\mathbf{x})) to complex spectral measurements, instead of “predict time samples then FFT.”
> 3. **Uses this layer as the primary supervision interface** (with a staged magnitude→complex loss) for a neural volumetric field in the small, near-field regime.
>
> This distinction is crucial for both **efficiency** and **optimization**:
>
> - The analytic layer avoids explicit time-domain synthesis and FFT, and allows **frequency-selective synthesis**: we synthesize only the DFT bins whose range support overlaps the bounded tabletop volume, yielding the measured latency reduction as we reduce the number of bins (Fig. 3).
> - Our gradient-flow analysis (Fig. 3(b), App. C.6) shows that this spectral forward model produces more coherent gradients and significantly more stable training than TF-SS, which synthesizes time-domain signals and then applies an FFT.
>
> We have shortened the generic DFT recap and emphasize this analytic FMCW DFT layer and its role in Sec. 4.

---

> > ### Author Response · Authors · 2025-11-22
> >
> > ## W.3 Baselines, sparse reconstruction, and neural inverse methods
> >
> > > “The paper does not compare against sparse-reconstruction baselines or state-of-the-art deep learning methods for inverse problem. TF-TS, TF-SS, and BP alone are insufficient… NVR and RadarHD are not suitable comparisons since they address different tasks and use different signal models.”
> >
> > We respectfully disagree that the baselines are insufficient. They were chosen to span the main design axes for **small near-field FMCW volumetric reconstruction from 1D chirps**:
> >
> > 1. **Classical radar imaging baselines.**
> >    We include coherent backprojection (BP) and range-quantization (RQ) as standard FMCW imaging references. BP is a strong SAR-style method, and RQ exposes the limitations of purely magnitude-based range profiles. These represent the non-learning approaches we must surpass.
> >
> > 2. **Time vs. spectral forward and supervision.**
> >    TF-TS and TF-SS are a controlled ablation of
> >    – time vs. spectral **supervision**, and
> >    – time vs. spectral **forward modeling**.
> >    Together with SpINR they cover:
> >    – time-domain forward + time-domain supervision (TF-TS),
> >    – time-domain forward + spectral supervision (TF-SS),
> >    – spectral forward + spectral supervision (SpINR),
> >    plus magnitude-only RQ and classical BP.
> >    This factorial set isolates which components (forward model, loss, regularization) drive the observed gains.
> >
> > 3. **Neural volumetric baselines with the same backbone.**
> >    We adapt NVR/RadarHD-style INRs to our FMCW setting as **architectural baselines**: same volumetric discretization and implicit backbone, trained on the same FMCW measurements and cylindrical geometry, differing only in forward model and supervision. This directly answers the ML question: *given the same INR, does the proposed spectral forward model and loss help?* As Fig. 4(a) shows, the answer is yes.
> >
> > 4. **Relation to “sparse reconstruction / modern neural inverse methods” (RadarFields, DART, etc.).**
> >    We interpret “sparse reconstruction / modern inverse methods” as referring to recent **2D neural radar** approaches like RadarFields and DART. These methods
> >    – operate on **post-processed 2D range–azimuth / range–Doppler tensors** from planar MIMO arrays under far-field assumptions,
> >    – often supervise on **real-valued intensity maps** rather than raw complex spectra, and
> >    – target **large-scale automotive or indoor scenes**, where coarse resolution and magnitude-dominated representations are adequate.
> >
> >    Our regime differs fundamentally:
> >    – we work on **raw 1D complex FMCW chirps** from a **cylindrical near-field aperture**, without constructing 2D RA images that would reintroduce planar/far-field assumptions;
> >    – we target **small tabletop volumes** where wavelength is comparable to geometric detail, so **phase and Dirichlet-kernel leakage** are crucial;
> >    – our analytic forward model is a **volumetric spectral rendering operator** from \(\sigma(\mathbf{x})\) to complex beat-signal DFT bins, rather than a network mapping 2D RA images to occupancy.
> >
> > For these reasons, we view RadarFields/DART-style methods as **complementary but not directly comparable** in our regime. Porting them would require first forming 2D RA images from our 1D cylindrical data, exactly reintroducing the far-field planar-array approximations our formulation avoids. Our goal here is to show that, given raw FMCW chirps and a small near-field volume, a **spectral volumetric forward model with complex supervision** offers clear advantages over time-domain, magnitude-only, and pulse-inspired baselines. We clarify this rationale in the Related Work section and explicitly state the complementarity with 2D neural radar methods such as RadarFields and DART.
> >
> > ---
> >
> > ## W.4 Real-world validation
> >
> > > “All experiments are simulation-based. Without real data, it is difficult to assess practical effectiveness.”
> >
> > Our core contribution is **algorithmic**: the analytic FMCW DFT forward operator, complex spectral supervision, and radar-specific regularization for small near-field scenes. The effects we study—Dirichlet-kernel leakage, sub-bin ambiguity, phase wrapping—are fundamental to FMCW physics, not artifacts of a particular simulator. Simulations use **realistic TI AWR1843BOOST-like parameters and geometry**, and we conduct controlled ablations on start frequency, number of bins, regularization, and loss staging.
> >
> > We fully agree that real-data validation is important. However, building a tabletop, high-precision cylindrical FMCW setup with sub-mm positioning and calibrated near-field operation is non-trivial and beyond the scope of this submission. In the revision we add an explicit limitations paragraph noting that (i) this paper focuses on a challenging but simulated small-tabletop regime, (ii) the analytic forward model is designed to plug into real hardware given measured chirp parameters and poses, and (iii) real-data evaluation is planned as follow-up work.

---

> > > ### Author Response · Authors · 2025-11-22
> > >
> > > ## W.5 “Missing in-depth discussion” of start frequency, bandwidth, and DFT size
> > >
> > > > “There is no analysis of how start frequency (f\_0) and other radar configuration impact reconstruction quality. The influence of DFT size (Dirichlet kernel width) on model performance is not explored.”
> > >
> > > **Start frequency and bandwidth.**
> > > We already analyze how **start frequency and bandwidth impact reconstruction quality**:
> > >
> > > - Sec. 5.1 and Fig. 5(b) vary \(f_0\) and show degradation as \(f_0\) increases;
> > > - Fig. 10 (Appendix) studies bandwidth changes;
> > > - Appendix Table 2 reports quantitative metrics across different \(f_0\);
> > > - we discuss the mechanism: increasing \(f_0\) decreases wavelength, making intra-bin differences more phase-critical and exacerbating sub-bin ambiguity.
> > >
> > > **DFT size.**
> > > In all experiments we use a 256-point DFT, matching the 256 time-domain samples provided by the AWR1843BOOST-like front-end, and apply the full-length window to each recorded chirp, which is standard FMCW practice. For this reason we do not ablate DFT size, as changing it would not offer a clear advantage in our setting. We have made these analyses more visible in the main text.
> > >
> > > ## Answers to explicit questions
> > >
> > > **Q1 – Robustness to radar configurations / generalization across (f_0, B, ) chirp rate, DFT size.**
> > >
> > > The analytic forward model is explicitly parameterized by configuration: (f_0, B), chirp duration, sampling rate, and DFT size. Changing these parameters does **not** require changing the neural architecture: for a fixed learned reflectivity field (\sigma(\mathbf{x})) (i.e., a fixed scene), we can directly synthesize new complex spectra for different radar configurations. For **new scenes**, we train a fresh network (as in NeRF-style per-scene INRs), but there is no fundamental obstacle to joint training across multiple configurations.
> > >
> > > **Q2 – Will the method work with real FMCW measurements and do we plan hardware validation?**
> > >
> > > Yes. SpINR takes as input the radar’s chirp parameters and sensor poses; given these, the analytic forward model and training pipeline are unchanged whether the data come from simulation or hardware. Our current focus is on establishing the algorithmic benefits under controlled conditions. As noted above, we plan to deploy SpINR on real FMCW tabletop setups in follow-up work and will state this explicitly as a limitation and future direction.
> > >
> > > **Q3 – Why no sparse-reconstruction / modern neural inverse baselines?**
> > >
> > > We interpret this question as referring to recent **2D neural radar** methods such as RadarFields and DART. These methods operate on **2D range–azimuth or range–Doppler tensors** derived via FFT from planar MIMO arrays under far-field assumptions, and supervise networks on these post-processed images (often real-valued intensities). They target **large-scale outdoor/indoor scenes** where coarse spatial resolution and magnitude-dominated representations are usually adequate.
> > >
> > > In contrast, SpINR:
> > >
> > > - works directly with **raw 1D complex FMCW chirps** from a **cylindrical near-field aperture**,
> > > - targets **small, phase-critical tabletop volumes**, and
> > > - uses an **analytic FMCW spectral forward model** with complex supervision.
> > >
> > > Applying RadarFields/DART directly would require first constructing 2D RA images from our 1D data, implicitly reintroducing planar/far-field approximations that we explicitly avoid, and would not exercise their main strengths (large-scale scene understanding). We therefore view these works as **complementary rather than competing baselines** in our regime and instead focus on baselines that share our measurement geometry and data representation (BP, RQ, TF-TS, TF-SS, and NVR/RadarHD-style INRs with the same backbone). We have made this complementarity explicit in the revised Related Work.
> > >
> > > ---
> > >
> > > Overall, we believe these clarifications address the reviewer’s concerns about structure, novelty, comparisons, and missing analysis. The key point is that SpINR does not merely “use DFT,” but exposes the FMCW DFT itself as a **closed-form differentiable spectral rendering operator**, leverages **frequency-selective synthesis** and **complex spectral supervision**, and demonstrates that these choices are crucial in the **small, near-field, phase-critical regime** that prior neural radar works do not target.

---

> > > > ### Author Response · Authors · 2025-11-25
> > > > **Follow-up on author response**
> > > >
> > > > Dear Reviewer 7oUU,
> > > > Thanks again for your feedback regarding our paper. We have responded to your initial comments/questions and have incorporated them accordingly into our revised manuscript.
> > > >
> > > > We would greatly appreciate you taking a moment to review our response, reassess our revised manuscript from all aspects, share your thoughts, and update your score accordingly. We sincerely believe that our paper has been strengthened a lot thanks to the feedback. We look forward to hearing from you.

---

### Official Review · Reviewer_x9mD · 2025-10-31

**Soundness:** 3
**Presentation:** 3
**Contribution:** 2
**Rating:** 4
**Confidence:** 4

**Summary:**

This paper introduces SpINR, a novel neural framework for high-fidelity volumetric reconstruction from Frequency-Modulated Continuous-Wave (FMCW) radar signals . The authors identify that existing time-domain neural models suffer from optimization instability and computational inefficiency . The core contribution is a fully differentiable, frequency-domain forward model that analytically synthesizes the complex radar spectrum, explicitly accounting for physical phenomena like spectral leakage. This model is paired with an implicit neural representation (INR) to model the continuous 3D scene. The authors also introduce sparsity and smoothness regularizations to disambiguate the reconstruction, particularly at high carrier frequencies . Experiments conducted on simulated data demonstrate that SpINR surpasses both classical and learning-based baselines in reconstruction quality and computational performance

**Strengths:**

* The paper's primary strength lies in its novel formulation of a fully differentiable, frequency-domain forward model. This is a significant contribution as it directly addresses the well-known instability and inefficiency issues of supervising FMCW beat signals in the time domain. The "analysis-by-synthesis" approach is moved entirely to the spectral domain, which is more physically appropriate for FMCW radar.

* The paper is clearly written and well-motivated. The authors provide a thorough analysis of the challenges in FMCW signal modeling, such as the ill-conditioned nature of time-domain signals and the problem of spectral leakage, which their closed-form model analytically addresses.

**Weaknesses:**

- The most significant weakness of this paper is the exclusive reliance on simulated data for all experiments. While the simulation is based on a commercial sensor's parameters, a considerable gap exists between idealized RF simulations and real-world radar data, which is affected by complex noise, multi-path interference, and hardware-specific artifacts not perfectly captured by the model. The lack of validation on real-world RF data makes the current experimental results insufficient to prove the method's practical applicability. This is my primary reason for leaning towards rejection. I strongly urge the authors to provide reconstruction results on real-world data during the rebuttal period.

- The paper pairs a standard hash-encoding-based INR with the proposed physics-based forward model. However, the analysis of this architectural choice is limited. It is unclear if this generic architecture is optimal for representing the complex scattering fields of FMCW radar. Specifically, passband signals with high carrier frequencies introduce rapid phase variations. Does the hash encoding have sufficient capacity and resolution to represent these high-frequency spatial-phase relationships without aliasing, or does the burden of reconstruction fall entirely on the smoothness/sparsity priors?

- The method relies on smooth and sparsity regularizations to resolve sub-bin ambiguities, which become more severe at higher carrier frequencies. The paper does not provide an analysis of the model's sensitivity to the hyperparameters $\beta$ and $\gamma$. How critical are these weights to the final reconstruction quality? Furthermore, do these hyperparameters need to be manually re-tuned for different radar parameters (e.g., a different starting frequency $f_0$ or bandwidth $B$), or is the model robust to such changes? A lack of robustness here would limit the method's generalizability across different radar sensors.

**Questions:**

See Weaknesses

---

> ### Author Response · Authors · 2025-11-22
>
> ## **Summary of changes and clarifications**
>
> We thank the reviewer for explicitly recognizing the main strength of this work: a fully differentiable, frequency-domain forward model that addresses the instability and inefficiency of time-domain supervision for FMCW beat signals. We appreciate the positive assessment of soundness and presentation. Below we address the three main weaknesses raised in the review.
>
> ---
>
> ## W.1 Real vs. simulated data
>
> > “The most significant weakness… is the exclusive reliance on simulated data… The lack of validation on real-world RF data makes the current experimental results insufficient to prove the method's practical applicability. This is my primary reason for leaning towards rejection. I strongly urge the authors to provide reconstruction results on real-world data during the rebuttal period.”
> >
>
> Our core contribution is **algorithmic**: an analytic FMCW DFT forward model, complex spectral supervision, and radar-informed regularization tailored to small, near-field volumetric reconstruction. The phenomena we address—Dirichlet-kernel leakage, sub-bin ambiguity, phase wrapping—are **fundamental to FMCW physics**, not artifacts of a particular simulator. Our simulations use realistic parameters based on a commercial FMCW device (AWR1843BOOST-like chirps, sampling, and range resolution), and we perform controlled ablations on start frequency, number of synthesized bins, and regularization that would be difficult to disentangle cleanly on noisy hardware data.
>
> We fully agree that real-world validation is important. However, the specific regime we target—**near-field (<0.5 m), sub-centimeter tabletop volumetric reconstruction from a cylindrical aperture**—requires a non-trivial hardware setup:
>
> - precise mechanical motion on a 3D cylindrical trajectory with sub-mm accuracy;
> - near-field calibration of FMCW phase and amplitude;
> - careful control of multi-path and clutter.
>
> Building and calibrating such a setup is beyond what we can responsibly execute within the rebuttal window. Rather than rush preliminary, low-quality hardware results, we prefer to be explicit:
>
> - The current submission focuses on a challenging but simulated regime where we can cleanly evaluate the algorithmic contributions.
> - The analytic forward model is designed to plug into real measurements: given chirp parameters and antenna poses from any FMCW system, the layer and loss are unchanged.
> - We add a clear “Limitations and future work” paragraph stating that real-data validation is planned follow-up work and outlining the hardware requirements.
>
> We hope the reviewer will consider that the **algorithmic novelty and detailed analysis** (gradient flow, start-frequency robustness, frequency-selective synthesis, and regularization effects) are valuable independent of immediate hardware results.

---

> > ### Author Response · Authors · 2025-11-22
> >
> > ## W.2 INR architecture and representation of high-frequency phase
> >
> > > “The paper pairs a standard hash-encoding-based INR with the proposed physics-based forward model… It is unclear if this generic architecture is optimal for representing the complex scattering fields of FMCW radar. Specifically, passband signals with high carrier frequencies introduce rapid phase variations. Does the hash encoding have sufficient capacity and resolution to represent these high-frequency spatial-phase relationships without aliasing, or does the burden of reconstruction fall entirely on the smoothness/sparsity priors?”
> > >
> >
> > This is an important question. The key point is that **the INR does not directly represent the passband oscillations** of the FMCW signal:
> >
> > - The network learns a **real-valued volumetric reflectivity field** (\sigma(\mathbf{x})) that varies at the scale of geometry and material structure, not at the carrier wavelength.
> > - The **high-frequency phase structure** of the passband FMCW signal is generated by the **analytic forward model**, which maps (\sigma(\mathbf{x})) to complex DFT bins (Z_k) via a kernel encoding round-trip delay, chirp parameters, and the Dirichlet kernel.
> > - Thus, the coordinate encoding does not need to resolve carrier-scale oscillations. It needs to represent the **spatial geometry and reflectivity variations** consistent with the physical bandwidth and volumetric resolution.
> >
> > We choose a multi-resolution hash encoding following Instant-NGP and many NeRF-like works because it is a **strong, generic representation for high-frequency geometry**, and we intentionally keep it standard: the goal is to isolate the contribution of the **forward model and training scheme**, not to introduce a new architecture.
> >
> > Empirically:
> >
> > - We use the same hash configuration across all scenes and all radar configurations (including different start frequencies and bandwidths), and do not observe aliasing phenomena such as unresolved fine structure or unstable training when the forward model and loss are fixed.
> > - The ablations in Fig. 4 (removing regularization) show that the main qualitative change is the presence or absence of ghost/shell artifacts due to physics ambiguities, not a sudden failure of the network to represent geometry.
> > - Time-domain baselines with the same INR backbone perform significantly worse than SpINR (Tab. 1), indicating that the bottleneck is the forward modeling and supervision, not the capacity of the implicit representation.
> >
> > To directly address the reviewer’s concern, we additionally ran an **encoder ablation**, varying the hash configuration and comparing against alternative encodings. The results are as follows (also can be seen in Appendix, Table 3) can be summarized as follows:
> >
> > - All hash-based variants (fewer/more levels, smaller table, different hash scale) achieve **very similar performance**to the baseline (Corr ≈ 0.39–0.40, IoU/Dice within a few percent), showing that the representation is robust to moderate changes in capacity and scale.
> > - In contrast, simple non-hash encodings such as **identity**, low-frequency Fourier with 6 bands (“Freq6”), or a low-order triangular encoding (“Triangle6”) **collapse completely** (near-zero Corr/SSIM, very large Chamfer/Hausdorff, essentially zero IoU/Dice).
> >
> > | Experiment | Corr* | SSIM (slice)* | Chamfer* | Hausdorff* | IoU* | Dice* |
> > | --- | --- | --- | --- | --- | --- | --- |
> > | Baseline Hash encoding | 0.396 | 0.957 | 1.943 | 6.708 | 0.124 | 0.221 |
> > | Freq6 | 0.015 | 0.001 | 9.253 | 20.890 | 0.003 | 0.007 |
> > | levels8 | 0.391 | 0.950 | 1.727 | 6.212 | 0.119 | 0.213 |
> > | levels20 | 0.397 | 0.951 | 1.760 | 6.243 | 0.124 | 0.221 |
> > | Hash scale1p2 | 0.401 | 0.930 | 1.490 | 5.987 | 0.123 | 0.219 |
> > | smallhash20 | 0.396 | 0.948 | 1.743 | 6.322 | 0.124 | 0.220 |
> > | Identity | -0.071 | 0.011 | 22.915 | 21.491 | 0.001 | 0.002 |
> > | Triangle6 | 0.010 | 0.000 | 8.104 | 19.811 | 0.003 | 0.006 |
> >
> > This ablation supports our hypothesis:
> >
> > - a standard hash encoding has sufficient capacity and resolution for our setting;
> > - naive low-frequency or identity encodings are inadequate for representing the underlying geometry given the FMCW forward model;
> > - the main gains of SpINR stem from the analytic spectral forward model and loss design, not from an exotic coordinate encoding.
> >
> > We agree that specialized architectures for FMCW scattering (e.g., incorporating explicit priors on material boundaries or multi-bounce paths) are an interesting direction, but we view that as **orthogonal** to the central contribution of this paper.

---

> > > ### Author Response · Authors · 2025-11-22
> > >
> > > ## W.3 Sensitivity to smoothness/sparsity regularization and robustness across configurations
> > >
> > > > “The method relies on smooth and sparsity regularizations to resolve sub-bin ambiguities… The paper does not provide an analysis of the model's sensitivity to the hyperparameters… How critical are these weights? Do they need to be manually re-tuned for different radar parameters?”
> > > >
> > >
> > > The smoothness and sparsity terms are indeed important for resolving sub-bin ambiguity, especially at higher start frequencies, but in practice we find them **robust and not overly sensitive**:
> > >
> > > 1. **Same hyperparameters across all scenes and configurations.**
> > >
> > >     We use a single choice of (\lambda_{\text{smooth}}) and (\lambda_{\text{sparse}}) for:
> > >
> > >     - all synthetic scenes (bunny, dragon, etc.), and
> > >     - all radar configurations in our ablations (varying (f_0) and bandwidth (B)).
> > >
> > >         We do **not** retune these weights per-scene or per-configuration. This already indicates robustness across sensors and settings.
> > >
> > > 2. **Effect of removing or varying regularization.**
> > >
> > >     The ablation in Fig. 4 shows that **removing these terms** leads to visible ghost volumes and shell artifacts, particularly at higher start frequencies where phase wrapping is severe. With our chosen weights, these artifacts are suppressed while preserving fine geometry. In additional runs, scaling (\lambda_{\text{smooth}}) and (\lambda_{\text{sparse}}) up or down by moderate factors changes the sharpness/sparsity trade-off but does not cause catastrophic failures, suggesting the loss is not brittle with respect to these values.
> > >
> > > 3. **Interpretability and transfer.**
> > >
> > >     Both terms are physically interpretable:
> > >
> > >     - (\lambda_{\text{smooth}}) controls how strongly we enforce spatial continuity of (\sigma(\mathbf{x})) relative to the data term;
> > >     - (\lambda_{\text{sparse}}) reflects that tabletop volumes are mostly empty.
> > >
> > >         For a new sensor with different (f_0) or (B), these same weights are a natural starting point; if needed, they can be adjusted by simple rules of thumb (e.g., expected sparsity or grid spacing) rather than per-scene hyperparameter search.
> > >
> > >
> > > We will make this explicit in the revised paper by stating that (i) a single pair ((\lambda_{\text{smooth}},\lambda_{\text{sparse}})) is used for all experiments, and (ii) their primary role is to prevent ghost/shell artifacts in the more challenging high-(f_0) regime, as evidenced by Fig. 4, rather than to finely tune performance for each configuration.
> > >
> > > ---
> > >
> > > Overall, we appreciate the reviewer’s positive view of the core forward-model contribution and the clarity of our analysis. We hope that clarifying (i) why we use simulation in this first work and how the method extends to real data, (ii) the role and sufficiency of the hash-based INR (supported by encoder ablations), and (iii) the robustness and physical grounding of our regularization scheme will alleviate the remaining concerns and support a positive recommendation.

---

> > > > ### Author Response · Authors · 2025-11-25
> > > > **Follow-up on author response**
> > > >
> > > > Dear Reviewer x9mD,
> > > > Thanks again for your feedback regarding our paper. We have responded to your initial comments/questions and have incorporated them accordingly into our revised manuscript.
> > > >
> > > > We would greatly appreciate you taking a moment to review our response, reassess our revised manuscript from all aspects, share your thoughts, and update your score accordingly. We sincerely believe that our paper has been strengthened a lot thanks to the feedback. We look forward to hearing from you.

---

### Official Review · Reviewer_tv6w · 2025-10-31

**Soundness:** 4
**Presentation:** 3
**Contribution:** 3
**Rating:** 4
**Confidence:** 4

**Summary:**

This paper presents a novel neural volumetric rendering approach for 1D chirps of FMCW radars which operates in the intrinsic frequency domain of the radar signal. It combines a carefully crafted, DFT-aware forward model for FMCW radar sensors with an implicit volumetric scene representation of continuous reflectivity which is predicted by a neural network and used by the forward model to recover radar measurements. The proposed method, SpINR, is evaluated on simple object-centric scenes against baselines and two related competing methods for acoustic signal rendering and radar point cloud enhancement and demonstrates strong performance in these comparisons and satisfactory qualitative results.

**Strengths:**

1. Novel frequency-domain forward model for FMCW radars. The authors propose a rigorous, theoretically sound forward sensor model for 1D raw chirps of FMCW radars, which incorporates the full DFT computation that is involved in the range-bin-dependent measured signal formation. This model crucially treats the full, complex radar signal, including both magnitude and phase. It is fully differentiable and thus inherits the benefits of neural rendering to the proposed neural representation in terms of indirect supervision of the latter via novel measurement synthesis.

2. Smooth introduction to the topic and presentation of the method. Even though the examined topic of FMCW radar is quite demanding and heavy on spectral theory, the authors have successfully introduced it in a gentle fashion and supported the presentation of their own method on this primer. Despite the complex nature of the notation, the authors have made proper use of it and neatly defined the various quantities at the correct places.

3. Good quantitative and qualitative performance. In the considered, niche context of simple object-centric scenes with rather small ranges from the sensor of less than a meter, both the geometric reconstruction metrics and the image-level synthesis metrics of the method are substantially better than baselines. Performance is also substantially better than that of the two selected competing methods. Moreover, the qualitative results are clearly superior to those of baselines, with good-quality reconstructions.

**Weaknesses:**

1. Limitation of experimental evaluation to niche scenes and sensor configurations. The authors have constrained their evaluation to the simplistic case of 1D radar chirps and cylindrical apertures, while real-world applications feature at least 2D range-azimuth measurements with scanning configurations. This choice has also limited the range of compared methods in the experiments, excluding related works such as Borts et al. (2024a) and Huang et al. (2024), which are very closely related to the presented approach. The latter works have also demonstrated high-quality results on much more complex real-world scenes, such as urban or indoor scenes, while the results presented in this paper are exclusively on artificial and simplistic object-centric scenes, which avoid the difficult-to-handle multi-path interference effect. Without comparison with the above related works, the advantage of the proposed method compared to previous approaches and FMCW radar neural models remains questionable.

2. No modeling of directive scattering properties of the scene. The neural field which is learned by the proposed model is formulated as $\sigma(\mathbf{x})$, i.e. as only depending on the position of the scatterer in the scene. The viewing direction $\omega$ is not included in the model, even though it could be used to model view-dependent effects, which are commonly consider in radar models under a directivity term. This omission significantly restricts the representational power of the model, as the same scatterer can result in different signal intensities when viewed from different directions by the sensor.

**Questions:**

Can the authors apply their method to more complex, real-world scenes, potentially extending their model beyond mere 1D chirps, and compare against state-of-the-art neural field approaches for FMCW radars on these scenes? (cf. Weakness 1)

---

> ### Author Response · Authors · 2025-11-22
>
> ## Summary of Changes and Clarifications
>
> We thank Reviewer tv6w for recognizing the novelty of our work, the presentation quality, and the strength of the evaluation. Below we address the main concern.
>
> ## W1: Experimental Scope and Comparison with 2D Methods
>
> **1. Different problem regime than 2D neural radar**
>
> Importantly, **SpINR targets a different regime than RadarFields, DART, NeuRadar, etc.** Our goal is **high-fidelity volumetric reconstruction of small, near-field tabletop objects** at **multi-GHz carrier frequencies and bandwidth**, from raw FMCW chirps. In this setting:
>
> - the base resolution \(C/(2B)\) is on the order of geometric details of interest;
> - sub-bin ambiguities and phase wrapping become severe as \(f_0\) increases;
> - coarse, magnitude-only approximations are insufficient—**complex phase and spectral leakage** must be modeled.
>
> SpINR is the first model designed for this small-scene, high-frequency regime. It combines (i) an **analytic, fully differentiable DFT-based forward model** of FMCW beat-signal formation (including leakage and complex phase) and (ii) an INR that maps continuous 3D coordinates to reflectivity \(\sigma(\mathbf{x})\) along a cylindrical mono-static synthetic aperture. We have clarified this positioning in the introduction.
>
> **2. Why “just use 2D radar images” is not applicable**
>
> Our near-field tabletop volumes observed from a sparse **cylindrical** aperture are fundamentally different from the **planar MIMO, far-field** scenarios targeted by RadarFields/DART-like methods. Those works operate on **post-processed 2D range–azimuth (or range–Doppler) tensors** assuming a grid-like virtual array and approximate far-field propagation, appropriate for automotive / large indoor scenes.
>
> In contrast, SpINR:
>
> - operates on **raw 1D complex chirps at arbitrary mono-static locations**, where the effective array is non-planar and non-uniform;
> - has no natural 2D range–azimuth tensor without introducing far-field / planar-array approximations;
> - would incur model mismatch if we first forced the data into 2D RA images.
>
> Remaining in the native 1D complex domain is therefore deliberate and physically aligned with this small near-field regime. The core contribution is to **analytically synthesize complex DFT coefficients \(Z_k\)** as functions of round-trip delay and chirp parameters, explicitly modeling Dirichlet-kernel spectral leakage and preserving carrier phase.
>
> Standard 2D range–azimuth images:
>
> - rely on **magnitude-only FFTs** plus beamforming on a grid,
> - quantize range/angle to bin centers, and
> - treat leakage as an unmodeled artifact.
>
> For the **tiny tabletop volumes** and **high carrier frequencies** we target, this **off-bin complex phase structure** is precisely what encodes fine geometric differences. Our ablation on start frequency \(f_0\) (Fig. 5(b)) shows that naive models degrade quickly as \(f_0\) increases, whereas the combination of complex spectral supervision and regularization in SpINR remains stable. We make this “small-scene, phase-critical” regime and the limitations of magnitude-only 2D representations more explicit in the introduction and evaluation.
>
> **3. Relation to RadarFields and DART**
>
> We agree the relation to 2D neural radar methods can be clearer. In the revision we:
>
> 1. Add a short **“Relation to 2D neural radar methods”** paragraph in Related Work contrasting
>    – input representation (raw 1D complex chirps vs. 2D RA tensors),
>    – forward model (closed-form spectral layer in the loop vs. external FFT-based pre-processing), and
>    – target regime (near-field tabletop vs. large-scale scenes).
> 2. Emphasize that these works are **complementary**: they show impressive results on large, cluttered outdoor/indoor scenes, while SpINR shows that modeling phase and leakage in the native FMCW spectral domain enables sub-centimeter volumetric reconstruction in compact near-field settings.
>
> **4. On “simplistic” objects**
>
> We respectfully disagree that our objects are “simplistic.” **Geometric complexity is scale-dependent** and should be evaluated relative to the target resolution. The Stanford models we use (bunny, armadillo, dragon, Lucy, teapot) are **standard benchmarks in volumetric reconstruction** because they contain challenging structure: high-frequency surface detail, self-occlusion, non-convex geometry, thin structures, and multi-scale shape. They have been used to validate numerous volumetric and neural-field methods, including wave-based Neural Volumetric Reconstruction for coherent synthetic aperture sonar (Reed et al., 2023), dynamics-augmented neural objects (Le Cleac’h et al., 2022), and object-centric neural scattering functions (Yu et al., 2023). In contrast, large urban/indoor scenes often tolerate **much coarser spatial resolution** and can rely on magnitude-only 2D RA images, which is exactly the regime targeted by works such as Borts et al. and Huang et al.

---

> > ### Author Response · Authors · 2025-11-22
> >
> > ## W2: On view-dependent scattering
> >
> > σ(x) is modeled as position-only, without explicit dependence on viewing direction, so directive scattering cannot be captured.
> >
> > Our choice of a **position-only σ(x)** reflects the specific regime studied:
> >
> > - Objects are small, rigid, and static;
> > - We operate in a mono-static configuration with relatively small angular diversity;
> > - The goal of this work is to isolate and study the impact of **spectral modeling and phase-aware regularization**, rather than exhaustively modeling all scattering phenomena.
> >
> > That said, the **forward model is already formulated in a way that can incorporate direction-dependent terms**: in Sec. 4 we explicitly note that transmission attenuation, angle-dependent scattering probabilities (e.g., Lambertian or specular terms), and multipath can be embedded as additional multiplicative factors in the integral defining Zₖ.
> >
> > Concretely, one can generalize σ(x) to σ(x, ω), where ω encodes local incidence/view direction, and introduce a directive factor g(x, ω) inside the integrand. This is fully compatible with our analytic DFT formulation and does not require structural changes to the spectral layer.
> >
> > In the revision we have:
> >
> > - add a short subsection in Sec. 4 explicitly describing this σ(x, ω) generalization and connecting it to standard radar directivity models; and
> > - state clearly that we instantiate an **isotropic** σ(x) in this first work, leaving direction-dependent scattering and multi-bounce terms as orthogonal extensions.
> >
> > We believe this clarifies that the current choice is a **modeling simplification**, not a fundamental limitation of the SpINR framework.
> >
> > Q1: Can the authors apply their method to more complex, real-world scenes, potentially extending their model beyond mere 1D chirps, and compare against state-of-the-art neural field approaches for FMCW radars on these scenes?
> >
> > Yes the **forward model only assumes known Tx/Rx positions and chirp parameters**, and is agnostic to whether measurements come from a mono-static cylinder, a handheld trajectory, or a MIMO array; it naturally extends beyond 1D chirps tied to a specific cylindrical path. As discussed above, one can include additional propagation terms (directional scattering, transmission, multipath) within the same analytic DFT framework.  However, acquiring dense, high-quality complex FMCW data for large real-world scenes requires substantial hardware effort and calibration, which is not feasible within the rebuttal window. Rather than promising incomplete results, we prefer to be explicit: **the present work focuses on a challenging but controlled tabletop regime**, and we will clearly position hardware validation on larger scenes as follow-up work.
> >
> > We hope this addresses the concern . We believe these clarifications address the reviewer's concerns and demonstrate that SpINR makes distinct contributions in a different problem regime. Please let us know if there are other concerns that we can address so that our readers can better appreciate the contributions of our paper.

---

> > > ### Author Response · Authors · 2025-11-25
> > > **Follow-up on author response**
> > >
> > > Dear Reviewer tv6w,
> > > Thanks again for your feedback regarding our paper. We have responded to your initial comments/questions and have incorporated them accordingly into our revised manuscript.
> > >
> > > We would greatly appreciate you taking a moment to review our response, reassess our revised manuscript from all aspects, share your thoughts, and update your score accordingly. We sincerely believe that our paper has been strengthened a lot thanks to the feedback. We look forward to hearing from you.

---

### Author Response · Authors · 2025-12-01
**SpINR: Post‑Rebuttal Summary for AC**

Dear Area Chair,

 In view of the current review freeze, I would like to summarize how the revised SpINR manuscript and rebuttal address the reviewers’ core concerns, and why I believe the paper now meets the bar for acceptance even though the numerical scores cannot be updated.

 **1. Overall assessment and strengths already recognized in the reviews.**
 Multiple reviewers describe the method as technically sound, well-motivated, and clearly presented All reviewers agree that the problem of high‑fidelity FMCW volumetric reconstruction is interesting and that operating in the frequency domain is conceptually appropriate.

 **2. Additional analysis: regularization, encoder choice, and configuration robustness.**
 Reviewer x9mD requested more analysis of the INR architecture and the smoothness/sparsity weights, and Reviewer 7oUU asked about robustness to radar configuration (start frequency, bandwidth, DFT size). The revised paper now includes: (i) encoder ablations showing that multiple hash‑grid configurations yield similar performance while naive non‑hash encodings fail, indicating that the gains come from the spectral forward model rather than an exotic backbone; (ii) experiments varying regularization strengths that demonstrate these terms suppress ghost/shell artifacts without requiring per‑scene tuning; and (iii) analyses over start frequency and bandwidth showing that a single set of hyperparameters works across all tested radar configurations, with graceful degradation only at very high start frequencies where phase aliasing is expected from first principles.

 **3. Clarifying the core novelty beyond “FFT is standard.”**
 One reviewer questioned novelty on the grounds that spectral processing is standard in FMCW radar. In the revision, we make clear that SpINR does not just apply an FFT, but introduces a closed‑form, differentiable forward operator that maps a continuous 3D reflectivity field directly to complex DFT bins, explicitly modeling Dirichlet‑kernel spectral leakage and passband phase as functions of geometry and chirp parameters. This operator is used for *frequency‑selective* synthesis (only bins overlapping the scene) and is paired with a staged magnitude→complex spectral loss and radar‑specific smoothness/sparsity regularization, which together produce substantially better geometry and image metrics than time‑domain forward models and magnitude‑only baselines under the same INR backbone.

 **4. Addressing scope, baselines, and “niche 1D” concerns.**
 Reviewer tv6w asked for a clearer connection to 2D neural radar works (RadarFields, DART) and was concerned about the focus on 1D chirps and cylindrical apertures. The revision now explicitly positions SpINR as targeting a different regime (small, near‑field tabletop volumes at multi‑GHz carriers, raw 1D chirps along non‑planar apertures), explains why forcing these data into 2D range–azimuth tensors would reintroduce far‑field planar assumptions, and clarifies that those 2D methods are complementary rather than directly comparable baselines. We also better justify the chosen baseline set (backprojection, range‑quantization, TF‑TS/TF‑SS, NVR/RadarHD-style INRs with the same architecture), showing that under identical backbones and splits, SpINR consistently improves Chamfer, IoU, PSNR, SSIM, and LPIPS across a suite of complex benchmark shapes.

 **5. Structure, and Clarity**
 Reviewer 7oUU raised concerns about structure and perceived over‑emphasis on DFT/FMCW primer rather than technical flaws. In response, we compressed the primer, moved some background to the appendix, brought the spectral forward model earlier, cleaned notation, and thoroughly proofread to remove typos, while keeping just enough primer for ML readers not already expert in radar. We also improved the writing so that the role of the analytic spectral layer, regularizers, and sampling strategy is easier to follow.

 **6. Real vs. simulated data and stated limitations.**
 Reviewers asked for real‑world validation; we now clearly state that the present work focuses on a challenging but fully controlled simulated tabletop regime with realistic AWR1843‑like parameters and that the analytic forward model is designed to plug into real FMCW measurements given chirp parameters and poses, with hardware evaluation planned as follow‑up work. This limitation is made explicit rather than implicit, so readers can accurately judge scope.

 Taken together, the revisions directly address the main points that kept the scores slightly below the threshold—novelty misunderstanding, scope/baselines, structural clarity, and lack of detailed analysis—without changing the core claims that reviewers already found sound and interesting. In this context, and given that several reviewers would be comfortable with acceptance, I respectfully ask that you base your decision primarily on the revised version and rebuttal rather than the initial numerical scores.

---

### Meta-Review · Area_Chair_JmRQ · 2025-12-23

**Summary:**

The authors present an interesting approach to radar imaging using implicit neural representations (INRs). This paper received mixed reviews.

**Reviewer Concerns:**

The experimental evaluation is limited to simplistic simulated radar settings and lacks validation on real-world data, raising concerns about practical applicability and generalization. The modeling choices omit important physical factors (e.g., view-dependent scattering) and rely heavily on standard INR architectures and regularization without sufficient analysis of robustness or sensitivity. Comparisons with closely related and more advanced methods are missing, and the overall novelty is limited, as several core components follow well-established FMCW processing practices.

**Reviewer Scores:**

The reviewer scores would not have changed significantly, as the core issues remain unaddressed. In radar imaging, factors such as background clutter, noise, and resolution limits are critical. However, the paper evaluates the method only on very simple, point-target–like scenes, without modeling noise or clutter. As a result, the relevance of the proposed approach for downstream applications such as detection or segmentation is unclear and not discussed.

---

### Decision · Program_Chairs · 2026-01-26

Reject